# Modelling transmission of Middle East respiratory syndrome coronavirus in camel populations and the potential impact of animal vaccination

Amy Dighe [1,2] ✉, Thibaut Jombart[1] & Neil Ferguson [1]

Outbreaks of Middle East respiratory syndrome coronavirus (MERS-CoV) in humans are driven by recurring zoonotic spillover from camels, leading to demand for camel vaccination. With two vaccine candidates shown to reduce infectiousness, there is a need to better understand transmission of MERS-CoV in camels and assess the potential impact of vaccination. To help address this, we used age-stratified seroprevalence data and a combination of modelling methodologies to estimate key epidemiological quantities including MERS-CoV transmissibility in camels and to estimate vaccine impact on infection incidence. Transmissibility was higher in West Asia ($R_O$ interquartile range 7-14) compared to Africa (3-5) and South Asia (2-3), highlighting the need for setting-specific vaccination strategies. Modelling suggested that even if the vaccine only reduced infectiousness rather than susceptibility to infection, vaccinating calves could achieve large reductions in incidence in moderate and high transmission settings, and interrupt transmission in low transmission settings, provided coverage was high (70-90%).

Middle East respiratory syndrome coronavirus (MERS-CoV) causes severe acute respiratory disease in humans, with an estimated infection fatality ratio of ~22%[1]. Although capable of spreading rapidly in hospital settings[2,3], MERS-CoV transmission is inefficient in the general community and recurrent outbreaks in humans are driven by repeated zoonotic spillover from dromedary camels (*Camelus dromedarius*)[4] from here on referred to as "camels". The role of camels in ongoing transmission has led to demand for camel vaccination as part of a set of interventions to avert human cases[5]. With two vaccine candidates shown to reduce viral shedding[6,7], there is a need to assess the potential impact of camel vaccination on transmission. This is hindered, however, by poor understanding of the epidemiology of MERS-CoV amongst camels and a lack of mathematical models of transmission dynamics in the animals.

Most camels show no outward signs of MERS-CoV infection, making assessing the epidemiology of the virus in the zoonotic reservoir challenging. Cross-sectional surveys testing for antibodies against MERS-CoV or viral RNA have demonstrated that as well as being endemic in camel populations in West Asia where autochthonous human MERS cases are reported, the virus is present in camels in parts of South Asia and circulates widely across Africa where the majority of the world's camels reside[8,9]. Little is known about transmission intensity and how this varies within the global camel population which is highly heterogeneous in terms of structure and husbandry practices. Only one study has estimated the annual force of infection, estimating it to be 0.4 in ranch populations and 0.1–1.0 in pastoral populations in Kenya[10]. Knowledge of immunity is also limited, but longitudinal studies have provided two important insights.

[1]MRC Centre for Global Infectious Disease Analysis, Jameel Institute, School of Public Health, Imperial College London, London, UK. [2]Johns Hopkins Bloomberg School of Public Health, Baltimore, MD, USA. ✉e-mail: amy.dighe@imperial.ac.uk

Firstly, studies in a small number of mother-calf pairs observed a wave of infection sweeping through calf populations after maternally-acquired antibodies (mAbs) waned over the first few months of life, suggesting protective mAbs may play a role in infection dynamics[11,12]. Secondly, studies have demonstrated reinfection of previously seropositive animals, and rapid reinfection of infected animals in high density settings such as markets and holding pens[6,13,14]. Unfortunately, longitudinal studies have been too short or small to reliably measure how immunity and calving might lead to seasonal variation in infection. Evidence from phylogenetic analysis suggests the risk of spillover is highest between April and July[4], but this is not reflected by the incidence of primary cases[15]. These knowledge gaps around transmission intensity and immunity, taken together with the lack of mathematical models of transmission in camels, impede the design of informed animal vaccination strategies, and the evaluation of their potential impact.

Here, we use published age-stratified seroprevalence data from camel populations across Africa, West Asia and South Asia to fit catalytic models of seroconversion and produce population specific estimates of MERS-CoV transmissibility in camels. We then introduce a stochastic, age-structured, dynamic transmission model of MERS-CoV in camels, which we use to estimate key, epidemiological quantities, including $R_O$, the Critical Community Size (CCS) and periodicity of infections, thereby providing insights into how controllable MERS-CoV may be in different camel populations. Finally, we use our model to simulate vaccination assuming different efficacy scenarios. We evaluate the potential impact of vaccination on transmission in camels, as well as the optimal age for vaccination. Alongside empirical studies, insights from dynamic models such as those developed here could contribute to informing an effective response to the zoonotic transmission of MERS-CoV.

## Results

### Transmissibility of MERS-CoV in camel populations

**Force of infection (FoI).** To estimate the FoI, we fitted catalytic models of seroconversion to age-stratified seroprevalence data extracted in our previous systematic review of MERS-CoV in camels[8], assuming the seroprevalence data was beta-binomially distributed. Catalytic models estimate the per-capita rate at which seronegative animals seroconvert. Whilst the reliability of estimates does not dependent on seropositivity being immunologically protective, such models are sensitive to assumptions about maternally acquired antibodies (mAbs) and seroreversion and so we fitted four different models to examine these factors. We found that allowing a proportion of calves to be born with protective mAbs in model 3 improved model fit, as did the inclusion of seroreversion due to waning of antibodies acquired following infection in model 2, though the additional improvement in fit afforded by seroreversion on top of mAbs in the best fitting model (model 4) was fairly small (Supplementary Table S1). We assumed test sensitivity and specificity were high for both neutralisation and non-neutralisation-based tests. The ranking of model fit was robust to the use of alternative assumptions where sensitivity of neutralisation tests was modelled to be lower at 85% (Supplementary Table S2). Model selection was also generally robust to exclusion of each dataset from the analysis, with the model with mAbs alone or with mAbs and seroreversion outperforming the others (Supplementary Fig. S2).

Parameters governing rates of antibody waning and data overdispersion were estimated as common to all studies, while the FoI was allowed to be study-specific. We estimated that mAbs waned rapidly in the first few months of life, lasting on average 2 months (95% credible interval (CrI): 1–4 months) in our best fitting model, but that antibodies wane slowly following infection, lasting ~17 years (95% CrI: 9–33 years) – approaching the lifespan of camels. However, it can be difficult to distinguish life-long antibodies from repeated boosting using catalytic models. Parameter estimates were largely robust to exclusion of each

data set with the exception of the data collected in Egypt which, when excluded, increased the estimated duration of mAbs to 4.8 months, similar to the 4.2 months estimated in model 3 (Supplementary Fig. S3). The overdispersion parameter, $k$, was estimated to be 2.5 (95% CrI: 2.0, 3.2), meaning the variance of the data was estimated to be around 3.5 times greater than what would be expected if the data were binomially distributed. When an uninformative prior for $k$ was used, the model tended to maximise $k$ meaning an extreme level of overdispersion could on its own account for the patterns observed in the data, irrespective of other epidemiological parameter values which were then unidentifiable. To circumvent this issue, a half normal prior with a standard deviation of 0.5 was used to constrain $k$ to values we believe to be more plausible.

The annual FoI of MERS-CoV in camels was generally higher in populations sampled in West Asia, and lower in those sampled in South Asia and Africa (Fig. 1, Table 1). This trend was consistent across all four models (Supplementary Table S3). The posterior mode for the FoI ranged from 0.1 to 3.0 across most study populations, except for in the population in UAE where seroprevalence was very high (>85%) in both calves and adults, perhaps indicative of a recent outbreak, and the FoI was estimated to be 7.1/year. Such high FoI estimates are necessary to explain the high seroprevalence measured in younger animals when assuming endemic transmission, but it is important to note that there is a risk of over-estimation of FoI for populations with seroprevalence approaching 100% since all high FoI values fit the data equally well. This effect is likely reflected by the long tails of some of the posterior distributions presented in Fig. 1A; we therefore chose to present the posterior mode as the central FoI estimates as we believe this to be a more representative than the mean. The best fitting model matched the data well, with model estimates overlapping the age-stratified seroprevalence data with only a few exceptions (Fig. 1B). The very high levels of seroprevalence measured in young calves in Tunisia and Kingdom of Saudi Arabia (KSA)[16] were underestimated, and the model struggled to reproduce the data collected in one study in KSA where seroprevalence was very high in calves and dropped considerably in adults[17], which was not seen in any other study.

**Basic reproduction number ($R_O$).** $R_O$ estimates can provide a more widely used intuitive measure of transmissibility which are more readily comparable with other diseases. We translated FoI into $R_O$ using a dynamic, age-stratified, stochastic model of MERS-CoV transmission (please see Methods for a detailed description of model assumptions). Central $R_O$ estimates ranged from 2 to 17 with the exception of one study in UAE in which almost all young calves were seropositive suggesting a very recent outbreak resulting in a very high estimate of FoI which translated to an $R_O$ of 34. Generally, $R_O$ estimates were higher for populations sampled in West Asia with an interquartile range of 7–14 across studies, compared to 3–5 in Africa and 2–3 in South Asia. These estimates were based on the common assumption that infectiousness is linearly related to viral load, which leads to an estimate of reinfections being 1% as infectious as primary infections based on shedding data from the control arm of the ChAdOx vaccine field study[6]. However, to illustrate the extent to which uncertainty around reinfection affects $R_O$ estimates, we also examined an extreme alternative where infectiousness is proportional to the logarithm of viral load leading to reinfections being 50% as infectious as primary infections (Table 1). Under this assumption, central $R_O$ estimates were lower than our central estimates, ranging from 1 to 4 overall, but still higher in West Asia than Africa and South Asia. $R_O$ estimates for populations with very high FoI estimates were most sensitive to assumptions about immunity as, in these populations, a higher proportion of infections at endemic equilibrium are reinfections and are therefore affected by relative infectiousness parameters. Neither varying the duration of complete immunity following infection, nor the relative susceptibility of previously infected individuals had a considerable effect on $R_O$ estimates

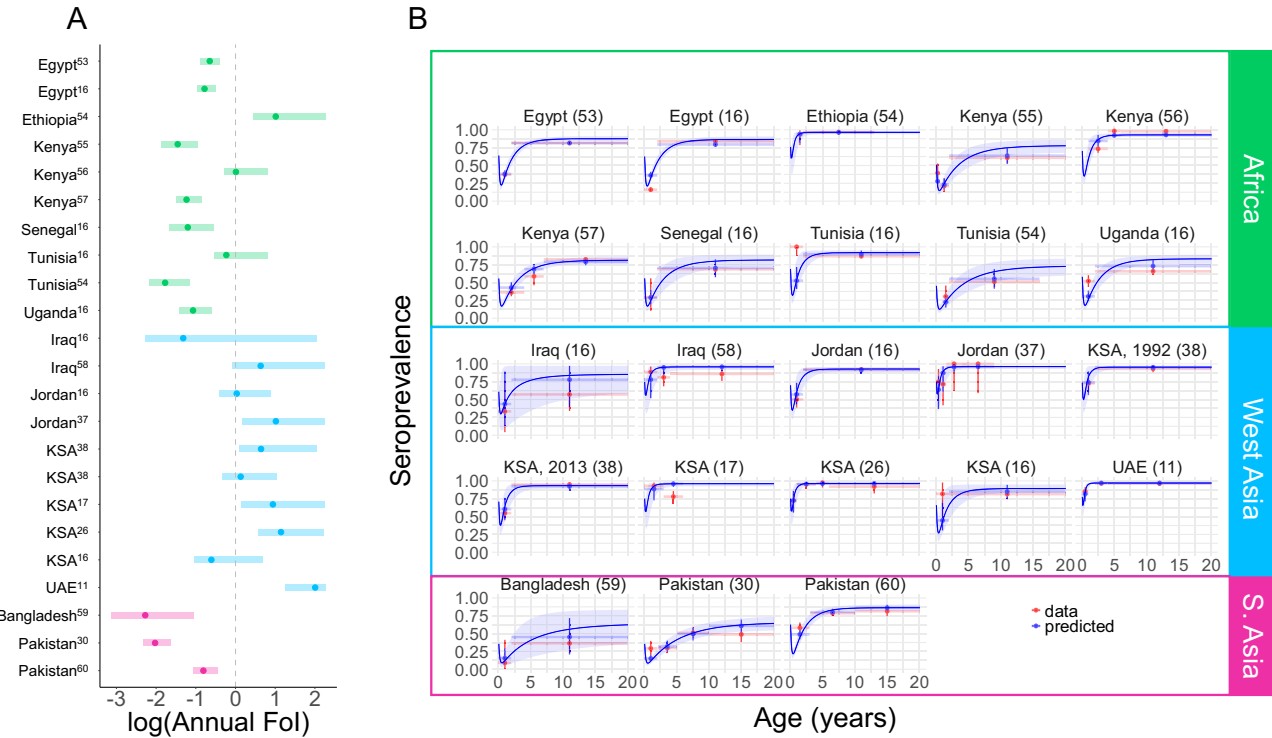

**Fig. 1 | Predicted seroprevalence and FoI estimates using Model 4. A** Posterior estimates for the FoI for each study, where the central point represents the posterior mode and the line represents the 95% CrI based on 10,000 samples of the posterior distribution. **B** The mean predicted seroprevalence by age (blue) with vertical 95% CrI based on 10,000 samples of the posterior distribution, fitted to age-stratified seroprevalence data (red) with points showing the proportion seropositive with vertical 95% CI (red) based on sample sizes varying from 6 to 1946 across age classes with individual *n* values presented in Table 1. Horizontal bars show age class width.

(Supplementary Table S4). When using estimates for the longer duration of mAbs (4.2 months) and for the FoI from the second-best fitting model without seroreversion, $R_O$ estimates were similar to those using best fitting model with seroreversion, albeit slightly lower (Supplementary Table S4).

### The critical community size (CCS)

We used the transmission model to estimate the population size above which the extinction of MERS-CoV transmission by chance becomes unlikely - the critical community size (CCS)[18]. We considered three different transmissibility levels spanning our $R_O$ estimates across different settings: low (South Asia and parts of Africa), moderate (Kenya and parts of West Asia, and high (Ethiopia and parts of West Asia) (Supplementary Fig. S4). The CCS was estimated to be well above reported herd sizes[19–21] and varied between ~10,000 and 70,000 camels depending on the transmissibility, seasonality of calving and underlying herd structure assumed in the transmission model (Table 2, Fig. 2). The CCS decreased as transmission intensity increased, except when births were highly seasonal and the population was modelled as homogeneous. In this case, high transmissibility resulted in a larger CCS than low or moderate transmissibility – likely because in this case, seasonal births and high $R_O$ drive explosive epidemics which nearly completely exhaust the susceptible population. Transmission could be sustained in smaller populations when births were less seasonally forced, and when the population was assumed to be homogenous as opposed to being structured into weakly connected patches intended to represent large herds or communities (Supplementary Fig. S5). Although sustained transmission was most dependent on total population size, the patch size did influence persistence. For low and moderate $R_O$ values, the greater the number of patches a given population was divided into, the less likely transmission was to be sustained. However, for high $R_O$ values, increasing the number of patches enhanced persistence (Supplementary Fig. S6) – likely due to the so-called "rescue effect", whereby, when transmission dies out in a local patch, infection may be reintroduced from neighbouring patches with asynchronous epidemics[22]. Under our alternative assumption that viral shedding is proportional to the log of infectiousness meaning past infection reduces infectiousness by only 50%, the CCS was smaller, with only 1000–30,000 camels needed to sustain transmission across depending on the transmission setting (Table 2, Fig. 2). Large commercial camel herds in West Asia have been reported to reach the lower end of this range[19].

### Periodicity of transmission

When births are assumed to follow seasonal patterns representative of those observed in KSA (see Methods), the number of infections over time has an annual periodicity in large populations, with peaks of a similar size occurring each year (Fig. 3A). In small populations, or when transmissibility is low, reflecting estimates for camel populations in South Asia and parts of Africa, biennial, triennial, and quadrennial periodicities - with patterns in the magnitude of annual peaks in infections repeating over 2-, 3- or 4-year cycles – are detected based on autocorrelation coefficients in a proportion of stochastic iterations (Fig. 3B). No seasonality in infections is observed when births are non-seasonal.

### The impact of vaccination

**Optimal target age.** We extended the transmission model to simulate the impact of age-targeted vaccination under multiple efficacy scenarios based on RNA shedding data from field studies and remaining uncertainties (Methods). The optimal target age for routine vaccination was assessed in a large population of camels so that overarching trends were not obscured by stochasticity. In our conservative scenario (scenario 1), in which vaccination reduces infectiousness of subsequent infections in all vaccinated animals but had no effect on

**Table 1 | Population specific estimates of the transmissibility of MERS-CoV in camels**

| | Dataset | Seroprevalence (%) | Test | FoI,λ | R₀ Relative infectiousness of reinfections | |
|---|---|---|---|---|---|---|
| | | | | Model 4: | 1% | 50% |
| Africa | Egypt[53] | <2 yrs 37% (n = 595)<br>≥2 yrs 82% (n = 1946) | MN | 0.5 (0.4, 0.7) | 4.2 (3.7, 5.1) | 1.9 (1.8, 2.1) |
| | Egypt[16] | <2 yrs 16% (n = 447)<br>≥2 yrs 84% (n = 1586) | MN | 0.5 (0.4, 0.6) | 4.0 (3.5, 4.7) | 1.9 (1.8, 2.0) |
| | Ethiopia[54] | 1–≤2 yrs 93% (n = 31)<br>2–13 yrs 97% (n = 157) | PM | 2.6 (1.5, 9.7) | 14.9 (9.6, 44.0) | 3.0 (2.6, 4.9) |
| | Kenya[55] | <6 m 39% (n = 61)<br>6m–2yrs 21% (n = 80)<br>>2 yrs 61% (n = 194) | PM | 0.2 (0.2, 0.4) | 2.6 (2.1, 3.5) | 1.7 (1.4, 1.8) |
| | Kenya[56] | 1–4 yrs 73% (n = 285)<br>4–6 yrs 98% (n = 116)<br>6 yrs 98% (n = 476) | ELISA | 1.0 (0.8, 2.3) | 6.8 (5.5, 13.4) | 2.3 (2.2, 2.9) |
| | Kenya[57] | <4 yrs 36% (n = 319)<br>>4 < 7 yrs 59% (n = 70)<br>>7 yrs 82% (n = 760) | ELISA | 0.3 (0.2, 0.4) | 2.9 (2.5, 3.7) | 1.7 (1.6, 1.9) |
| | Senegal[16] | <2 yrs 29% (n = 17)<br>≥2 yrs 69% (n = 181) | MN | 0.3 (0.2, 0.6) | 2.9 (2.3, 4.6) | 1.7 (1.5, 2.0) |
| | Tunisia[16] | <2 yrs 100% (n = 28)<br>≥2 yrs 87% (n = 754) | MN | 0.8 (0.6, 2.2) | 5.8 (4.6, 12.8) | 2.2 (2.0, 2.8) |
| | Tunisia[54] | <2 yrs 30% (n = 46)<br>≥2 yrs 54% (n = 158) | PM | 0.2 (0.1, 0.3) | 2.2 (1.8, 3.1) | 1.5 (1.3, 1.7) |
| | Uganda[16] | <2 yrs 52% (n = 150)<br>≥2 yrs 66% (n = 350) | MN | 0.3 (0.2, 0.6) | 3.2 (2.6, 4.5) | 1.8 (1.7, 2.0) |
| West Asia | Iraq[16] | <2 yrs 33% (n = 6)<br>≥2 yrs 57% (n = 21) | MN | 0.2 (0.1, 7.5) | 2.6 (1.7, 35.3) | 1.7 (1.3, 4.3) |
| | Iraq[58] | <2 yrs 89% (n = 44)<br>2–4 yrs 81% (n = 58)<br>>4 yrs 86% (n = 78) | ELISA | 1.8 (0.9, 9.4) | 10.6 (6.4, 43.0) | 2.7 (2.3, 4.8) |
| | Jordan[16] | <2 yrs 50% (n = 82)<br>≥2 yrs 92% (n = 222) | MN | 1.0 (0.7, 2.4) | 6.9 (5.0, 13.8) | 2.3 (2.1, 2.9) |
| | Jordan[37] | ≤2 yrs 74% (n = 31)<br>>2 yrs 100% (n = 14) | ELISA | 2.6 (1.2, 9.5) | 14.8 (7.8, 43.1) | 3.0 (2.4, 4.8) |
| | KSA[38] 1992-2010 | ≤2 yrs 55% (n = 104)<br>>2 yrs 95% (n = 98) | ELISA | 1.9 (1.1, 7.5) | 10.9 (7.2, 36.3) | 2.7 (2.4, 4.4) |
| | KSA[38] 2013 | ≤2 yrs 73% (n = 77)<br>>2 yrs 93% (n = 187) | ELISA | 1.2 (0.7, 2.8) | 7.5 (5.3, 15.7) | 2.4 (2.1, 3.0) |
| | KSA[17] | 1–2 yrs 93% (n = 71)<br>3–5 yrs 78% (n = 100) | ELISA | 2.3 (1.2, 9.5) | 13.2 (7.5, 43.1) | 2.9 (2.4, 4.8) |
| | KSA[26] | <1 yr 72% (n = 65)<br>1–3 yrs 95% (n = 106)<br>4–5 yrs 97% (n = 76)<br>>5 yrs 92% (n = 63) | ppNT | 3.2 (1.8, 9.1) | 16.6 (10.4, 41.7) | 3.1 (2.6, 4.7) |
| | KSA[16] | <2 yrs 82% (n = 11)<br>≥2 yrs 82 % (n = 211) | MN | 0.6 (0.4, 2.2) | 4.5 (3.4, 12.6) | 2.0 (1.8, 2.8) |
| | UAE[11] | ≤1 yr 85% (n = 108)<br>2–4 yrs 97% (n = 340)<br>>4 yrs 96% (n = 310) | ELISA | 7.1 (3.5, 9.8) | 33.7 (18.5, 44.5) | 4.2 (3.2, 4.9) |
| South Asia | Bangladesh[59] | <2 yrs 9% (n = 11)<br>≥2 yrs 36% (n = 44) | ppNT | 0.1 (0.0, 0.3) | 1.7 (1.3, 3.2) | 1.3 (1.1, 1.8) |
| | Pakistan[30] | ≤2 yrs 29% (n = 89)<br>2.1–5 yrs 30% (n = 208)<br>5.1–10 yrs 51% (n = 180)<br>>10 yrs 49% (n = 88) | ELISA | 0.1 (0.1, 0.2) | 1.9 (1.7, 2.3) | 1.4 (1.3, 1.5) |
| | Pakistan[60] | ≤3 yrs 58% (n = 177)<br>3.1–10 yrs 79% (n = 712)<br>>10 yrs 81% (n = 161) | ELISA then MN | 0.4 (0.3, 0.6) | 3.9 (3.3, 4.9) | 1.9 (1.8, 2.1) |
| Global | Rate of waning mAbs, ω | | | 4.9 (3.2, 9.6) | | |
| | Rate of waning Abs, σ | | | 0.06 (0.03, 0.11) | | |
| | Overdispersion, k | | | 2.5 (2.0, 3.2) | | |

MN microneutralisation test, ppNT pseudo particle neutralisation test, PM protein micro-array, ELISA Enzyme linked immunosorbent assay.

**Table 2 | The coverage necessary to interrupt transmission at the population level in two modelled populations**

| $R_O$ | $1/\rho$ | Vaccine coverage (%) needed to interrupt transmission in a population of: | |
|---|---|---|---|
| | | 75,000 split into patches of 3000 | 2,000,000 split into patches of 80,000 |
| 3.5 | 1 | 40 | NA |
| | 3 | 40 | 90 |
| | 10 | 40 | 70 |
| 7.0 | 1 | 80 | NA |
| | 3 | 60 | 100 |
| | 10 | 60 | 90 |
| 14.0 | 1 | 80 | NA |
| | 3 | 70 | NA |
| | 10 | 60 | 100 |

susceptibility, vaccination led to the greatest reduction in infection incidence when calves were targeted in the first few months of life (Fig. 4A). In the absence of vaccination, most animals were first infected when they were <1-yr old across all modelled transmission settings. In targeting younger calves, vaccination precedes first infection in a greater number of individuals, reducing their subsequent infectiousness. The reduction in incidence afforded by targeting younger animals was larger in higher transmission settings where first infections occurred earlier, with reductions in incidence diminishing quicker as the target age class was increased compared to in lower transmission settings. When the duration of vaccine induced effects was assumed to be relatively short and transmission intensity was moderate or high, vaccinating very young calves shifted the average time to first infection into older age groups, leading to a small increase in the annual incidence in adult animals of up to 10 per 1000 animals under our central model assumptions (Supplementary Fig. S7). Vaccinating at 6 months allowed large reductions in overall incidence of infection without seeing considerable shifting of first infections into adult animals. Note that adult incidence is considered here given it may be a proxy for zoonotic spillover risk to humans.

Under our optimistic scenario (scenario 2) in which we assumed vaccination reduced both infectiousness and susceptibility in all vaccinated animals, we saw the same pattern as in scenario 1, with greatest impact achieved by vaccinating in the first few months of life, and no notable increase in adult incidence when targeting 6-month-olds. Vaccinating older age groups after most first infections had occurred had almost no effect on incidence in scenario 1, whereas a slight reduction was still achieved by reducing the susceptibility of the older animals to reinfection in scenario 2 (Supplementary Fig. S8). In our third scenario, in which vaccination was only effective as a booster for previously infected animals with no impact in naïve animals, the optimal age for vaccination was early adulthood but even then, the reduction in incidence was minimal at <8% across all transmission intensities (Supplementary Fig. S8). The optimal target age for vaccination was robust to our different assumptions about the relationship between viral load and infectiousness.

**The impact of vaccination on transmission.** To explore the characteristics of the MERS-CoV vaccine and the vaccine coverage that would be necessary to achieve considerable reductions in infection incidence among camels, we simulated the impact of vaccination of 6-month-old calves in two modelled populations: one of 2 million camels comparable in size to that of KSA; and one of 75,000 camels comparable to that of a small camel-keeping Kenyan county.

In a population of 2 million camels divided into large homogenous patches, assuming the vaccine reduces infectiousness but not susceptibility, the vaccination coverage required to half the total

incidence over the 10 years following introduction was between 50 and 90% in 6-month-olds, depending on the duration of vaccine induced effects and the transmission intensity (Fig. 4B). When vaccine induced effects were long lasting, 50% coverage was required to half incidence in low transmission intensity settings, rising to ~80% coverage needed in high transmission intensity settings (Fig. 4B). When effects lasted 3 years, coverage of 70% was needed in low transmission settings rising to 90% when transmission intensity was high, and when vaccine induced effects only lasted one year, incidence could not be halved under any modelled setting. Alternatively, in a population of 75,000 camels, stochastic effects amplified the impact of vaccination: a coverage of <=50% in 6-month-olds could half total incidence in the 10 years following vaccine in low transmission intensity settings, even if effects only lasted 1 year. In a moderate transmission intensity setting, between 50 and 70% coverage was needed, and in high transmission intensity settings 70–90% coverage, depending on duration of vaccine induced effects. Across all transmission intensities, assuming the vaccine reduced susceptibility of vaccinated animals to 50% or 75% (efficacy scenario 2) only afforded a very small (~1% on average) additional reduction in incidence compared to when assuming the vaccine reduced infectiousness alone (Supplementary Fig. S9).

In the population of 2 million, when $R_O$ was low, coverage was high, and the effects of the vaccine were long-lasting, vaccination was capable of interrupting transmission and led to stochastic fadeout. In these cases, the difference in incidence between stochastic runs was often large (the 2.5% and 97.5% quantiles are represented by transparent ribbons in Fig. 4B). In low and moderate intensity settings, vaccination was able to interrupt transmission when coverage in 6-month-olds was very high, and the vaccine induced effects lasted at least 3 years (Table 2). In high transmission intensity settings transmission was only interrupted when coverage was 100% and vaccine induced effects lasted 10 years. In the smaller population of 75,000 divided into homogenous patches of 3000, stochastic fadeout occurred at lower coverages and across a wider range of scenarios. Vaccination was capable of reliably interrupting transmission when coverage ranged from 40 to 80% depending on transmission intensity and duration of vaccine induced effects. These estimates are based on mortality rates that reflect the current dominant camel husbandry systems described in Eastern Africa and Arabian Peninsula in which calves experience a high mortality rate largely driven by the slaughter of young males and then surviving adults (mostly females) experience a lower mortality rate. If meat production becomes mostly intensive in the future, characterised by large dense farms with rapid turnover of animals, we would expect a higher vaccine coverage to be necessary to interrupt transmission in these settings.

## Discussion

Understanding the transmission dynamics of MERS-CoV in camels is vital to evaluating the potential public health impacts of animal vaccination but has been hindered by the scarcity of data describing what is largely an asymptomatic infection in this species. By using age-stratified seroprevalence and viral load data extracted from published studies, we estimated the transmissibility of MERS-CoV in camels and developed a dynamic model of transmission, allowing for the first evaluations of the potential impact of camel vaccination under different efficacy scenarios. Whilst uncertainty around immunity and aspects of vaccine efficacy remains, we have gained several insights into the transmission dynamics and controllability of MERS-CoV in camels.

The transmissibility of MERS-CoV was generally estimated to be higher in camel populations in West Asia compared to those sampled in South Asia and Africa. All viruses sampled from camels in Africa have been classified into Clade C based on their genetic similarities. Strikingly, despite the large number of live camels imported from Africa, all viruses isolated from camels and humans in the Arabian Peninsula have belonged to genetically distinct clade A and B viruses – even those

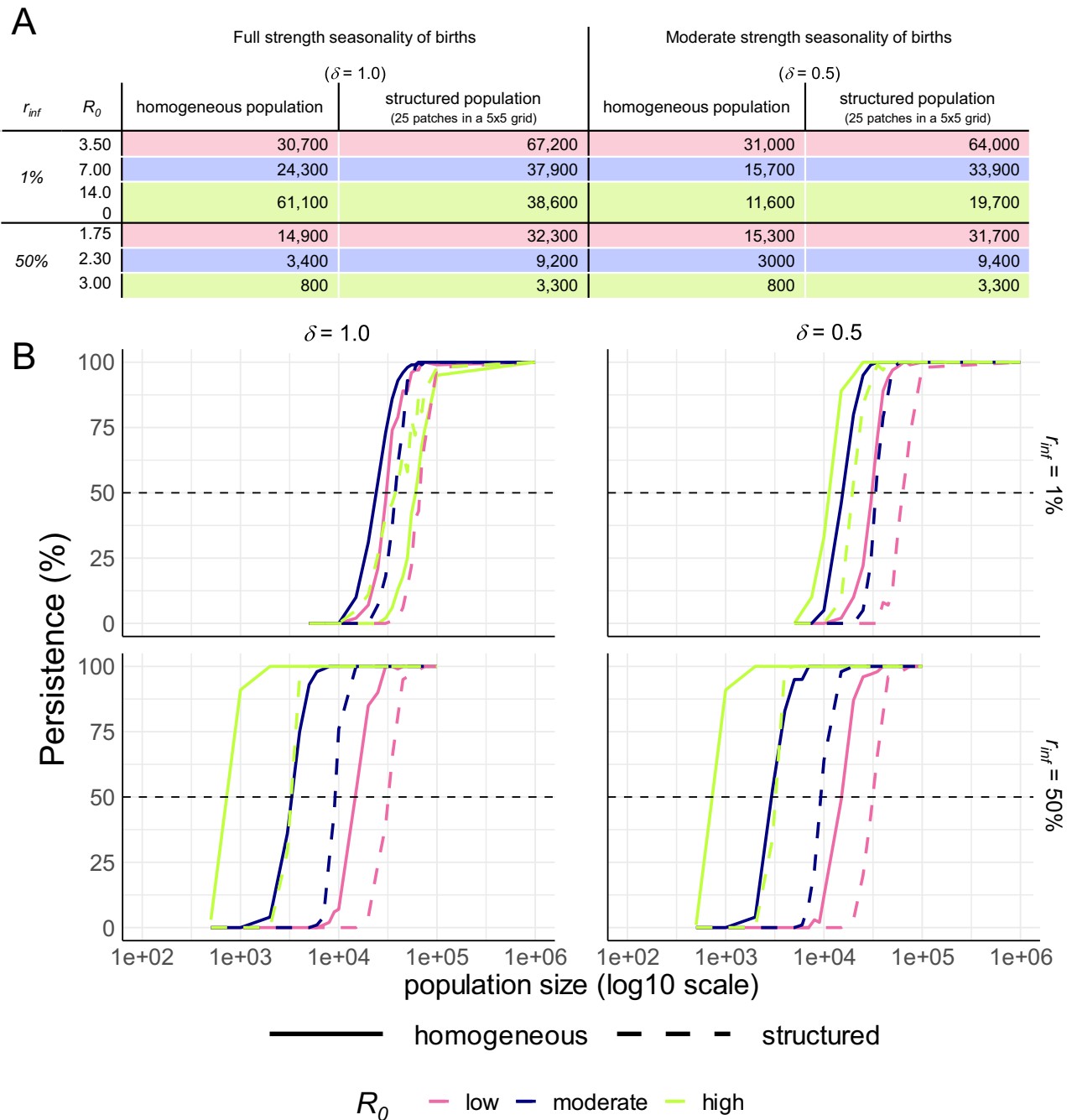

| $r_{inf}$ | $R_0$ | Full strength seasonality of births ($\delta = 1.0$) | | Moderate strength seasonality of births ($\delta = 0.5$) | |
|---|---|---|---|---|---|
| | | homogeneous population | structured population (25 patches in a 5x5 grid) | homogeneous population | structured population (25 patches in a 5x5 grid) |
| 1% | 3.50 | 30,700 | 67,200 | 31,000 | 64,000 |
| | 7.00 | 24,300 | 37,900 | 15,700 | 33,900 |
| | 14.00 | 61,100 | 38,600 | 11,600 | 19,700 |
| 50% | 1.75 | 14,900 | 32,300 | 15,300 | 31,700 |
| | 2.30 | 3,400 | 9,200 | 3000 | 9,400 |
| | 3.00 | 800 | 3,300 | 800 | 3,300 |

**Fig. 2 | The estimated critical community size (CCS) of MERS-CoV in camels.**
**A** The CCS is presented under two alternative values for the relative infectiousness of reinfected animals ($r_{inf}$), in different transmission settings ($R_0$). Estimates are shown for models assuming a homogeneous population of perfectly mixed animals and a structured population in which animals have more contact with those in their "patch" and weaker contact with those in surrounding patches. Estimates are shown assuming births follow seasonal patterns as strong as those observed in KSA ($\delta = 1$) and alternatively with a weaker seasonality ($\delta = 0.5$). **B** The percentage of stochastic model runs in which transmission persists by population size is shown under each scenario presented in panel (**A**), including different transmission intensities (line colour), and population structure assumptions (homogeneous = solid lines and structured population = dashed lines). Panels separate scenarios with differing strength of seasonality of births (columns), and relative infectiousness of reinfected individuals (rows). The horizontal dashed line indicates persistence in 50% of model runs - the point at which persistence becomes more likely than fadeout.

isolated from newly imported animals[23]. Our estimates of higher MERS-CoV transmissibility in camels in West Asia align with the results of a recent study that found clade C to have a reproductive disadvantage compared with clade A and B in human lung tissue[24], suggesting that the clade C viruses prevalent in Africa may be intrinsically less transmissible to humans, and perhaps between camels as well. However, by underpinning interactions between susceptible and infectious animals, variation in global camel husbandry practices could also potentially explain differences in transmissibility. The camel population is highly heterogeneous in terms of husbandry practice even at the local scale[19,20]. In the Arabian Peninsula, camel farming has become increasingly intensive and urban in the past 60 years, whilst remaining largely extensive pastoralist techniques elsewhere[25]. Further investigation of what is driving perceived differences between transmission intensities will be important for devising context-specific vaccination or other control strategies.

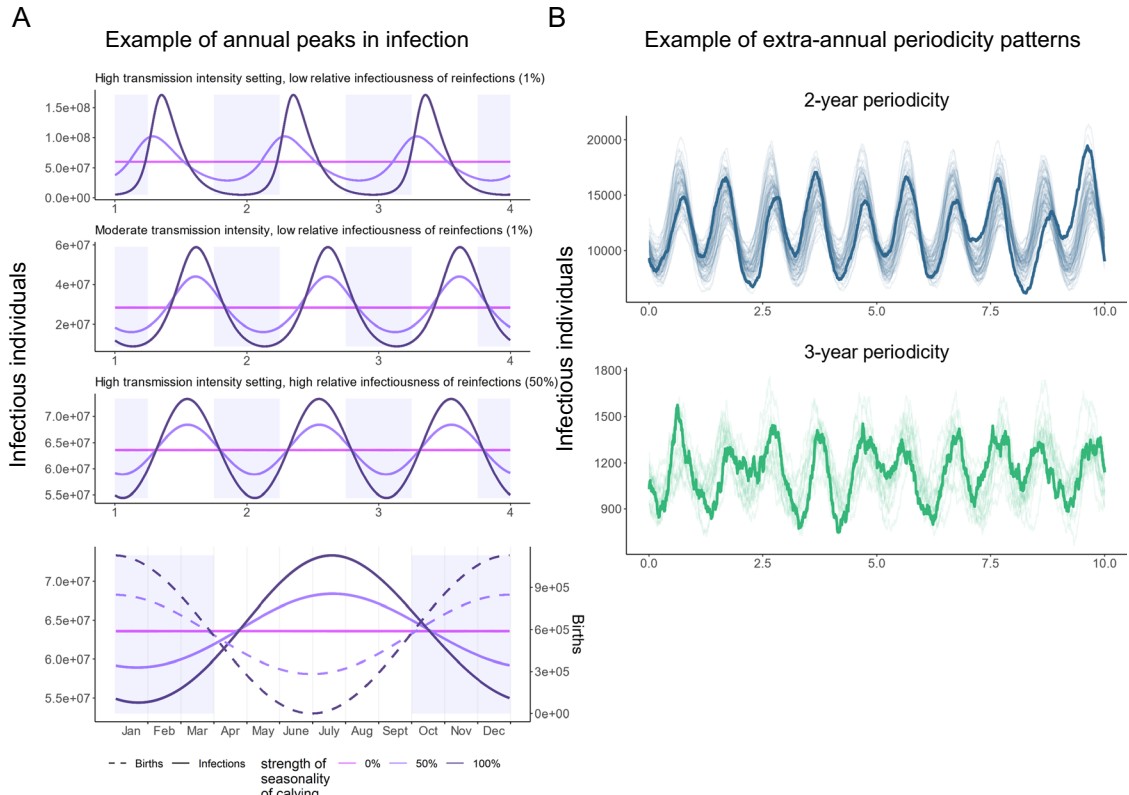

**Fig. 3 | The periodicity of simulated MERS-CoV epidemics in camels. A** Examples of simulated epidemic curves under three sets of conditions in which infections peak annually, shown in relation to the calving season (shown shaded in blue, with simulated number of births over the course of a year under different strengths of seasonality represented as dotted lines). A large population size was modelled for clarity. **B** Examples of stochastic epidemic curves with biennial (blue) and triennial (green) periodicity arising in smaller populations, with a single time series of each type highlighted in bold.

There are several limitations affecting our estimates of MERS-CoV transmission intensity. First, our FoI and $R_O$ estimates are based on age stratified seroprevalence surveys and the camel populations sampled are likely to be biassed towards countries with sufficient resources to detect human cases and to study MERS-CoV in their camel populations. It was not possible to estimate the transmissibility of MERS-CoV in Somalia and Sudan – the most camel dense areas in the world – due to a lack of age stratified seroprevalence data for these populations. Second, surveys used different tests to determine seropositivity, likely with different sensitivities and specificities. Since the FoI is estimated using relative differences in seroprevalence within a single study where a single test type was used, our estimates should not be greatly affected by this except when seroprevalence is very high and approaches the limit of sensitivity. To test the influence of differences in test sensitivity and specificity across test types, models were re-fit to the data whilst assuming reduced specificity of ELISAs and reduced sensitivity of neutralisation tests. Our ranking of FoI estimates, with generally higher estimates in West Asia and lower in Africa and South Asia, was robust to this change. Third, without high resolution data on camel population density and movements, we were not able to meaningfully explore potential density-dependence of transmission which could contribute to some of the variation in $R_O$. Detailed data describing local camel populations could be used to tailor our models to specific settings and account for such dynamics. Fourth, the rate of seroreversion could not be reliably identified due to uncertainty around test sensitivity and challenges distinguishing long-lasting antibodies from repeated boosting of antibodies following recurring infection in the catalytic model framework. Documented reinfection of seropositive animals in high density market pens shows that antibodies may not be a proxy for complete immunity to MERS-CoV. With this in mind, the rate of seroreversion was not used to inform

parameterisation of waning immunity in the transmission model. Instead, several alternative values were used to test sensitivity of vaccine impact estimates to this uncertainty. Last, we note that $R_O$ estimates are sensitive to uncertainty about the relative infectiousness of reinfections. Estimates are highest (and thus vaccination has the least impact on transmission) when infectiousness is assumed to be linearly related to viral load, leading to low infectiousness of reinfected animals compared to primary infections.

Our estimates of the CCS were larger than most reported herd sizes, which tend to be well under 1000 animals[19–21], emphasising the importance of focusing interventions on reducing inter-herd infections for interrupting transmission. The dependence of the CCS on transmissibility, together with the difference in $R_O$ across populations, suggests that MERS-CoV may be able to persist in a population 2–20 times smaller in high transmission settings found in the Arabian Peninsula, compared to lower transmission settings. Our simulations suggest that seasonality of births can be expected to drive annual, seasonal peaks in infection in large populations with the high transmission intensities we estimated for parts of West Asia, and Ethiopia. Whilst infections tended to peak outside of the calving season, the lag between the simulated peak in births and annual peak in infection depended on $R_O$. If MERS-CoV infections peak annually in some settings as these simulations suggest, there may be seasons in which risk of zoonotic transmission is elevated. Better understanding of when peaks occur would provide opportunities to mitigate risk and avert human cases. Although we considered seasonal calving, other factors such as annual migrations and events that affect camel mixing could also affect the transmission dynamics of MERS-CoV in the zoonotic reservoir. Ultimately, it is necessary to undertake long-term surveillance over several years to better ascertain the seasonality of MERS-CoV in camels.

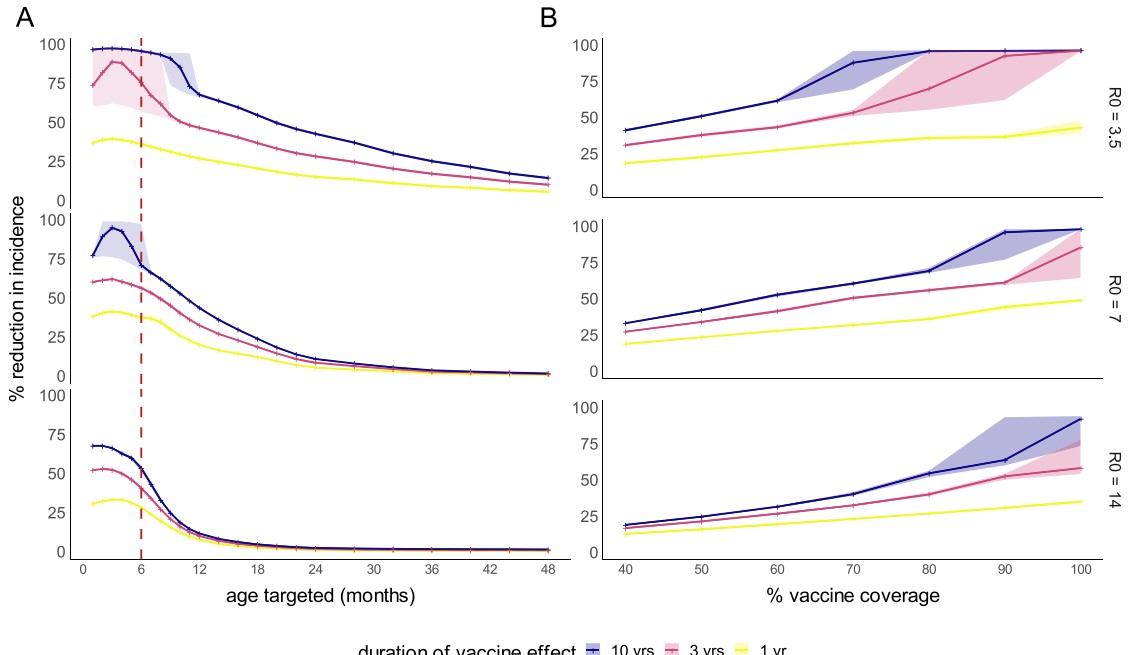

duration of vaccine effect ■ 10 yrs ■ 3 yrs ■ 1 yr

**Fig. 4 | The percentage reduction in incidence of MERS-CoV infection in camels by age and vaccine coverage. A** The percentage reduction in incidence of MERS-CoV infection in camels depending on age group targeted, in different transmission settings (rows), for varying durations of vaccine effects (colours). Transparent ribbons show the 2.5–97.5% quantiles (across 1000 stochastic model runs), with stochastic fadeout of transmission leading to larger differences across stochastic runs in some scenarios. The red dashed line indicates the 6-month age class. Vaccine coverage was assumed to be 80% in the target age class. **B** The percentage reduction in incidence of MERS-CoV infection in the 10 years following introduction of vaccination of 6-month-old calves, in camel populations of 2 million (comparable to that of KSA) made up of 25 homogenous patches of 80,000 animals connected to their nearest patches. Transparent ribbons show the 2.5–97.5% quantiles across 1000 stochastic model runs.

Our simulations suggest that if a MERS-CoV vaccine is able to reduce infectiousness in naïve and previously infected camels, routine vaccination could achieve large reductions in incidence, provided that a high proportion of calves are vaccinated. Little impact was seen if vaccine was only effective in previously infected animals. Although the ChAdOx1 MERS vaccine was measured to have poor efficacy in naïve animals in an initial field trial, potentially due to the animals' age[6], the MVA vaccine has been shown to reduce shedding in naïve animals[7]. Assuming independence of efficacy on age, we saw that vaccinating calves in their first few months of life maximises reductions in overall incidence among camels. Our observation that vaccination of very young calves led to more infections in adult animals in some scenarios highlights the importance of understanding age dependency in human-camel contact patterns across different populations. Vaccination strategies should be evaluated not only on their likely impact on transmission between camels, but also on how changes to the age distribution of infections in camels might affect zoonotic spillover to humans.

When coverage was high, our analysis indicted that vaccination can cause large reductions in infection incidence even for a vaccine assumed only to reduce infectiousness rather than susceptibility. When infectiousness was assumed to be proportional to viral RNA shedding, the coverage needed to interrupt transmission in a population of 2 million animals was over 70% across all scenarios, reaching 90–100% in moderate-high transmission intensity settings. This suggests that, in a large population with high levels of camel mixing and high MERS-CoV transmission intensity (such as some settings in West Asia) it would be difficult to entirely interrupt transmission through vaccination of calves alone, but that incidence could still be greatly reduced. It should be noted that vaccination of dromedaries is not likely to be used in isolation but rather considered as part of a suite of complimentary control measures in the animal population. However, to meaningfully consider interventions such as biosecurity measures or potential changes to husbandry practices in this modelling analysis

would require much more detailed data on current husbandry practices and population structures.

Our estimates of the potential impact of dromedary vaccination are limited by the absence of data on population structure and movement/trading patterns. Therefore, our modelling used a simple grid of connected sub-populations to approximate the structuring of the population into herds or geographic patches. In smaller or more fragmented populations with less mixing, our analysis indicates that interruption of transmission could likely be achieved with lower vaccine coverage than in more connected, larger populations. Tailoring our model to specific populations using data on camel herd sizes and animal movement would be necessary to provide more precise estimates of vaccination impact. The relationship between viral RNA shedding data and infectiousness also affects our estimates of vaccine impact. If rather than being proportional to viral load, infectiousness is proportional to log viral load, then vaccination would be expected to cause smaller decreases in infectiousness than for our central scenario and our simulations suggest that large reductions in infection incidence would be more difficult to achieve and/or require even higher vaccine coverage. In addition, vaccine effectiveness in naïve animals and the extent to which vaccination protects against infection were not clear from current field trials, so we included a range of scenarios as sensitivity analyses. As these data gaps are filled, it will be possible to improve mathematical models of MERS-CoV transmission in camels and become increasingly confident that they accurately represent transmission in the zoonotic reservoir.

Previous modelling studies of MERS-CoV have focused on human-to-human transmission. However, as recurring camel-to-human transmission drives human cases, there was a growing need for a model of transmission in the zoonotic reservoir. The model presented here provides a framework in which to simulate MERS-CoV vaccination strategies in camels which, together with improved data on camel mixing patterns and further empirical studies of vaccine efficacy, could

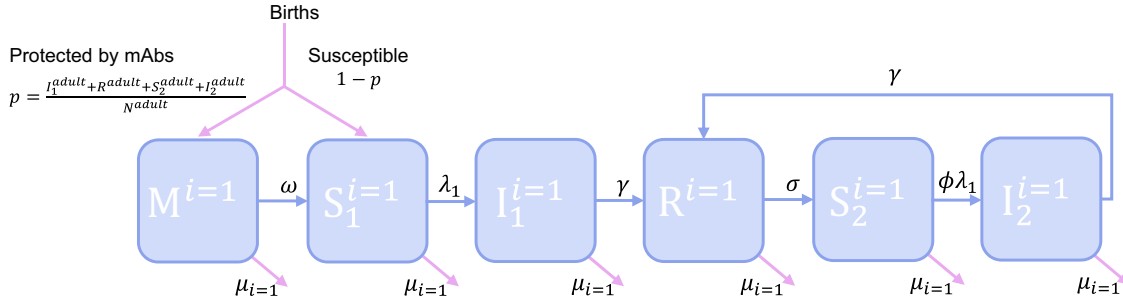

**Fig. 5 | A dynamic model of MERS-CoV transmission in camel populations.** A schematic of the model is shown for the youngest of i age classes (i = 1). Animals are born either protected by mAbs or susceptible, with a probability $p$, of being protected proportional to the fraction of adults previously infected at the time of birth.

The states represented are defined as follows M - maternally acquired immunity, $S_1$ - susceptible, $I_1$ - first-time infected, R - recovered, $S_2$ – susceptible again following infection and $I_2$ - reinfected. This structure is repeated for each of the further 48 age classes. The symbols used to represent rates are defined in Table 3.

inform effective responses to the zoonotic transmission of MERS-CoV. Efforts to better define the relationship between the number of infectious camels and the risk of zoonotic spillover events would allow the expected reduction in infection amongst camels to be translated into the expected number of human cases averted, permitting evaluation of the cost-effectiveness of camel vaccination as an intervention against human cases of MERS-CoV.

## Methods
Further details of methods are given in the Supplementary Materials.

### Estimating the transmissibility of MERS-CoV in camels
We estimated two different measures of the transmissibility of MERS-CoV in camels: the Force of Infection (FoI, $\lambda$) defined as the rate at which susceptible animals become infected, and the reproduction number ($R_O$) defined as mean number of individuals infected by a single infected individual in an entirely susceptible population.

**FoI.** We fitted catalytic models of seroconversion to age-stratified seroprevalence estimates from across Africa, South Asia and West Asia collated previously through a systematic review[8]. Since the catalytic modelling approach assumes seroprevalence estimates are derived from a random cross-sectional sample of individuals, we excluded 3 of the 19 reviewed studies based on their sampling strategies (please see *Supplementary Materials* for more details on study inclusion). To make the geographical range of the FoI estimates as comprehensive as possible, we used seroprevalence measures from one additional study published after the systematic review. This allowed us to include camel populations in Senegal and Uganda which were not previously represented in the literature[16]. The tests used to determine seropositivity varied between studies and included both neutralisation tests (NTs) and non-neutralising Enzyme Linked Immunosorbent Assays (ELISAs) (Table 1). NTs are shown to be highly specific to MERS-CoV antibodies with little cross-reactivity with other camel coronaviruses[26–29]. MERS-CoV IgG ELISAs have been measured to be 99% specific when correlated with NTs[30,31]. Whilst ELISAs are considered more sensitive than NTs as they can pick up non-neutralising antibodies[32], the seroprevalence measured by studies using NTs often approaches or reaches 100% in adult camels suggesting that – assuming they are indeed highly specific – they must also be highly sensitive. Therefore, we assumed a high sensitivity (98% for both test types) and specificity (99.5% for NTs and 98.5% for non-NTs) in our central results. We then conducted a sensitivity analysis assuming NTs to have a lower sensitivity of 85%.

We compared the fit of four models of seroconversion. In model 1 we assumed that all animals are born seronegative and become seropositive at a constant rate $\lambda$, as originally conceptualised by Muench

and now regularly applied to epidemiological data[33,34]. Since MERS-CoV reinfection has been documented in camels[6,13,14], in model 2 we extended model 1 to allow for seroreversion - with protective antibodies waning at rate $\sigma$. In model 3 we extended model 1 to allow a proportion of calves to be born seropositive due to protective mAbs which wane at rate $\omega$, as evidence suggests that calves born to seropositive mothers are shown to acquire MERS-CoV specific mAbs through colostrum[11,12]. Finally, our fourth model allowed for both mAbs and seroreversion. Please see the Supplementary Materials for equations describing the solutions used for each model. We fit the models within a Bayesian framework using Hamiltonian Monte Carlo (HMC) sampling algorithm implemented in the R software package rstan version 2.32.7[35]. Whilst we estimated the FoI per study to account for potential true differences between the FoI across husbandry systems, we assumed antibody waning rates to be constant, estimating them globally across all the datasets. We assumed that the seroprevalence data was beta-binomially distributed and reparameterised the beta-binomial distribution in terms of the mean probability of being seropositive and the overdispersion parameter $k$ where $k > 0$ and a $k$ approaching zero would indicate negligible overdispersion. A detailed reparameterization available in the Supplementary Material. In order to evaluate which of the models was best supported by the data, we compared their fit using the Deviance Information Criterion[36].

**The reproduction number ($R_O$).** We estimated $R_O$ of MERS-CoV in each study population by calibrating a dynamic model of MERS-CoV transmission (see next section) to the modal FoI, by varying the transmission intensity parameter, $\beta$, under different potential immunity scenarios. $R_O$ was approximated as the product of $\beta$ and the infectious period, $\gamma$. The one-to-one relationship between $\beta$ and the FoI meant that the credible intervals (CrIs) around the FoI could be used to propagate the uncertainty into the $RO$ estimates.

### Development of a dynamic model of MERS-CoV transmission in camels
**Infection.** Based on what we know about camel demography from the literature, and our estimates of transmissibility and maternal antibody waning, we developed a stochastic, age-structured model of MERS-CoV transmission in camels. The model structure is represented schematically in Fig. 5, with a single age class shown for clarity. All symbols used are defined in Table 3 alongside the parameter values and their sources. Camels are born either entirely susceptible to MERS-CoV infection (state $S_1$) or with complete protection by mAbs (state M) which wanes at a rate $\omega$ with calves becoming susceptible after an exponentially distributed period with a mean of ~2 months as estimated from the age-stratified seroprevalence data. The proportion of

## Table 3 | Transmission model parameters

| | Description | Values | Source |
|---|---|---|---|
| $N_O$ | Initial population size | Varied (50–10,000,000) | NA |
| $\bar{\alpha}$ | Mean birth rate | Varied annually around a mean of 0.000565 camel$^{-1}$ day$^{-1}$ based on initial population size. | Estimates of 45.2% annual fecundity in KSA[46] taken together with assumptions that 90% of the population are female due to high male removal rate[19] and that 50% of the female population are of reproductive age[19] |
| $\beta$ | Effective contact rate | 0.1–1.0 camel$^{-1}$ day$^{-1}$ | Calibrated to our FoI estimates from age-stratified seroprevalence data |
| $\gamma$ | Rate of recovery from infection | 1/14 days | |
| $\delta$ | Strength of seasonality of births | 1 (0, 0.5 also considered) | 19 see "Births" in Methods. |
| $\sigma$ | Rate of waning of complete immunity following infection | 1/30 days, 1/90 days (scenario with no complete immunity also considered) | NA |
| $\lambda_1$ | Rate at which susceptible animals become infected, equal to $\beta\frac{I_1+rI_2}{N}$ | 0.1–3.0 calibrated by varying $\beta$ | Our estimates from age-stratified seroprevalence data |
| $\omega$ | Rate of waning of mAbs | 0.0136 day$^{-1}$ | Our estimates from age-stratified seroprevalence data |
| $\mu_1$ | Daily mortality rate of camels <=2 yrs | 0.0011 camel$^{-1}$ day$^{-1}$ | Within the ranges described in ref. 46 but exact value set to balance mean birth-rate |
| $\mu_2$ | Daily mortality rate of camels aged >2 yrs | 0.00036 camel$^{-1}$ day$^{-1}$ | Within the ranges described in ref. 46 but exact value set to balance mean birth-rate |
| $\varphi$ | Susceptibility to reinfection relative to first infection | 0.75 (0–1 considered) | NA |
| $r$ | Infectiousness of reinfections relative to first infections | 0.01, 0.50 | 6 see "Immunity" in Methods. |
| $p$ | Probability of being born with mAbs | the fraction of adults >4 yrs old who have been infected previously at the time of the birth | NA |

calves born in state M is dictated by the proportion of animals of reproductive age (>4 years) which have been previously infected. Animals in $S_2$ become infected and transition to state $I_1$ with the FoI, $\lambda_1$, defined as the product of the effective contact rate, $\beta$, and the proportion of individuals in the population which are infectious:

$$\lambda_1 = \beta\frac{I_1+rI_2}{N} \qquad (1)$$

where $I_2$ is the number of reinfected individuals, $r < 1$ and represents the relative infectiousness of reinfections compared to first infections, and $\beta$ is varied to calibrate $\lambda_1$ to our FoI estimates from age-stratified seroprevalence data. We chose to model transmission as frequency-dependent rather than density-dependent. In many countries camel farming is still overwhelmingly extensive[25], with herds living and moving within large areas. In such settings, transmission is likely to depend more on the proportion of individuals infected rather than the absolute number. Modelling frequency dependence also enabled us to consider transmission at a wide range of scales and population sizes without requiring detailed data on animal stocking densities. To do this in a meaningful way whilst incorporating density-dependence would have required additional data on the relative sizes or densities of camel farms, grazing areas and markets and movement between these settings that is not currently available. No data is available on the potential latent period following MERS-CoV infection in camels. Infected animals are assumed to be instantaneously infectious. The period spent in state $I_1$ is exponentially distributed around a mean of 14 days in agreement with the duration of shedding reported in longitudinal studies[6,11,13].

**Immunity.** Whilst our inference from age-stratified seroprevalence suggests that under catalytic model assumptions antibodies may be long-lasting following infection, documented reinfection of seropositive animals and rapid reinfection in high transmission intensity environments indicates that MERS-CoV seropositivity is not a good proxy for protective immunity in camels[6,13,14]. We therefore explored multiple reinfection scenarios (Table 3). Following a short period of complete immunity in state R, individuals become susceptible to

reinfection in state $S_2$. Most animals found to be shedding MERS-CoV in field surveys are calves and naïve animals, suggesting there is some long-term protection offered by past infection[20,37–39]. To reflect this, the degree of susceptibility in state $S_2$ is modelled to be less than that experienced by individuals in state $S_1$, meaning individuals in state $S_2$ experience a reduced FoI, $\varphi\lambda_1$. Reinfected individuals in $I_2$ are modelled to be less infectious than individuals in $I_1$. This is based on measures of viral load collected in the control arm of the ChadOx1 MERS vaccine field study in camels[6]. We digitally extracted the daily mean viral load for seronegative calves (which we assumed to be infected for the first time during the study) and seropositive calves (which we assumed to be reinfected during the study) in the unvaccinated control group in from Fig. 4A of the online publication using PlotDigitizer version 2.2[40]. We then calculated the difference between the area under the viral load curve for each of the two groups. Reinfected animals were approximately 1% as infectious as first-time infected animals when assuming a linear relationship between viral load and infectiousness. The relationship between viral load and infectiousness is not well characterised. A trial of the MVA-based vaccine candidate in camels measured a similar decrease in a measure of infectious viral particles and a measure of viral RNA shedding, following vaccination of four calves[7]. However, the study was not designed to have the power to reliably define the relationship between infectious virus particles and infectiousness. Therefore, whilst our central results assume a linear relationship, with reinfected individuals 1% as infectious as first-time infections, we include a sensitivity analysis assuming that the relationship between viral load and infectiousness is logarithmic, with a relative infectiousness of 50% for reinfected animals.

**Age structure.** Inclusion of age structure is vital given the strong dependence of infection status and seroprevalence on age, as well as for simulating age targeted interventions. Fine age structure is especially important up until the age of four years to enable accurate representation of age within the window where first infections are happening and accurate, age-targeted intervention modelling. For this reason, the model is stratified into month-wide classes, with camels moving to the next age-strata every 30 days in a 360-day year. From the 48th month

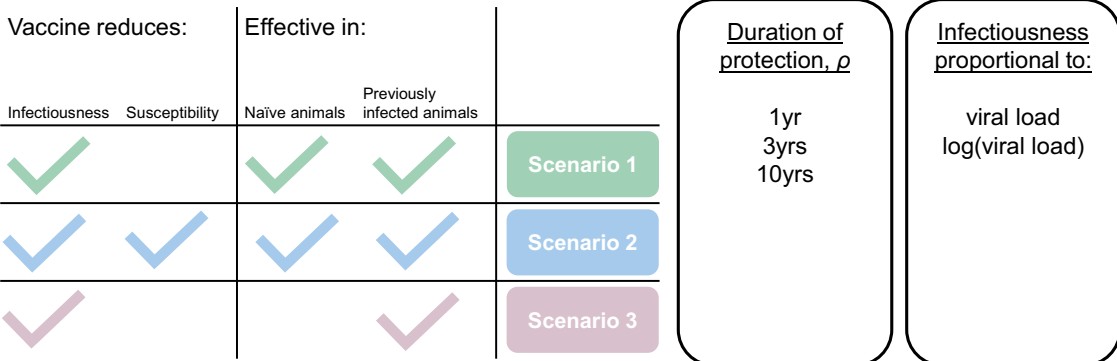

**Fig. 6 | Modelled vaccine efficacy scenarios.**

wide class, camels enter a class aged >4 years where they remain until death.

**Births.** Camel calving is reported to be strongly seasonal[19,41–45]. Studies in KSA report most calves being born between October and March, with one study quantifying this at 83% during the high season[19]. The calving season is very similar in Egypt where it is reported between October and April[45] and in Nigeria where surveyed pastoralists identified the calving peak to occur in the early dry season between October to December[41]. To capture this seasonality, the number of births per day is drawn from a Poisson distribution with a mean of $\alpha N_0$ where $\alpha$ varies annually as a function of cosine (Eqs. (2) and (3)) and $N_0$ is the initial population size. The strength of seasonality can be weakened by setting $\delta < 1$ during sensitivity analyses. However, when $\delta = 1$, 82% of births fall between October and March which is in line with the 83% reported for camel births in Qassim, KSA[19].

$$births \sim Pois\left(\alpha(t) * N_0\right) \tag{2}$$

$$\alpha(t) = \overline{\alpha\left\{1 + \cos\left(\frac{2\pi t}{360}\right)\right\}} \tag{3}$$

**Deaths.** Since MERS-CoV causes very mild disease in camels, infection is modelled to have no bearing on mortality. Camels die off from each disease state compartment at the same age dependent average rate $\mu_i$, with the mean number of deaths per day being equal to the size of the compartment multiplied by $1 - e^{-\mu_i}$. The model assumes a higher probability of calf death in the first two years of life than in adulthood, as reported in KSA[19,46]. The modelled mortality rates are calibrated to the birth rate to give a stable population size and are equivalent to ~40% mortality in the first two years of life and ~12% afterwards, similar to overall mortality estimates for populations in KSA which are described in the literature as 10–26%, depending on herd type[46].

**Structure.** For large populations, it becomes unrealistic to assume populations are well mixed. For example, in the population of ~10,000 camels in Laikipia county, Kenya[47], an individual camel is far more likely to have contact with individuals in its own herd or grazing area than with animals in other areas of the county. The movements and interactions between herds of camels are not well documented. To explore the effect population structure has on dynamics, we developed a rudimentary structured population model where subpopulations or patches are arranged over a grid (Supplementary Fig. S5). Individuals are most likely to be in contact with other individuals in the same patch, less likely to meet individuals in neighbouring patches, and do not meet individuals in distant patches. Until better data on population structure allows a more accurate representation of

networks and movements of camels within a region, the grid serves as a naïve representation of this reality.

We coded the model in R[48] version 3.5.3, using the package odin version 1.5.11[49] and ran stochastic iterations using odin.dust version 0.3.13[50,51].

**Estimating the Critical Community Size (CCS)**

To evaluate the CCS of MERS-CoV in camel populations, we estimated the size of the population required for transmission to be sustained for at least 25 years in a closed population with no external sources of infection. The CCS was defined as the population size at which transmission was sustained in at least 50% of stochastic model runs. We ran the model using population sizes ranging from 500 to 1,000,000 and estimated the precise population at which 50% persistence was achieved using linear interpolation. For our central results we fixed the number of patches at 25, varying the patch size to represent different total population sizes. We conducted a sensitivity analysis to understand the effect patch size had on transmission persistence varying the number of patches from 9 (a 3 × 3 grid) to 49 (a 7 × 7 grid).

**Evaluating the periodicity of infections**

To determine the average time between peaks in infections we estimated the autocorrelation between each simulated time series of infections and lagged versions of itself using Pearson's correlation test implemented through the acf function in the R "stats" package. The lag that maximised the autocorrelation coefficient was used to estimate the periodicity, for example if the lag that maximised the autocorrelation coefficient was between 350 and 370 periodicity was classified as annual. Very short lags of <100 days and any acf below the significance level using 95% confidence interval (CI) were excluded.

**Estimating vaccine impact**

We extended the transmission model to simulate vaccination by duplicating the set of disease states to create a parallel set of vaccinated states. Although two vaccine candidates have been shown to reduce viral shedding in camels, uncertainty remains around their ability to reduce susceptibility and around the effectiveness of the ChAdOx1 MERS vaccine in naïve animals. Due to these uncertainties, three main scenarios are modelled (Fig. 6). In our central scenario 1, the vaccine reduces infectiousness but *not* susceptibility to infection for all vaccinated animals. This scenario reflects the finding that all previously naïve vaccinated animals became infected when challenged. Challenge doses administered intranasally or by confinement with multiple infectious animals could be much higher than the average natural exposure, and 1/5 of the previously infected ChAdOx1 vaccinated animal did not become infected despite challenge so we also explored an alternative scenario 2 in which the vaccine reduces both infectiousness and susceptibility for

**Table 4 | Parameters used to simulate vaccination under different efficacy scenarios**

| | Description | Values | Source |
|---|---|---|---|
| $r_v$ | Relative infectiousness of vaccinated animals | 1% (50% as sensitivity analysis assuming infectiousness is proportional to the logarithm of viral load). | 6,7 |
| $r_{inf\_v}$ | Relative infectiousness of vaccinated previously infected animals | 0.15% (33% as sensitivity analysis assuming infectiousness is proportional to the logarithm of viral load). | 6 |
| $\varphi_v$ | Relative susceptibility of vaccinated animals | 100% (or 75% in scenario 2) | NA. Larger reduction in susceptibility unlikely given all vaccinated previously seronegative animals were infected in refs. 6,7. |
| $\varphi_{inf\_v}$ | Relative susceptibility of vaccinated previously infected animals | 75% (or 75% in scenario 2) | NA. Indication of reduced susceptibility[6] but unable to reliably quantify in small population |
| $1/\rho$ | The rate of waning of vaccine-induced effects | 1, 3 or 10 years$^{-1}$ | |

all vaccinated animals. Finally, we explored a third scenario in which the vaccine reduces both infectiousness and susceptibility but only in animals that have been previously infected. Although the MVA study measured a large reduction in infectiousness of previously seronegative vaccinated animals, the ChadOx1 vaccine was only measured to reduce shedding in previously infected vaccinated animals. Authors suggest the low efficacy in this group could be due to the naive animals being very young, but their age was comparable with those used in the MVA study. Parameters used in vaccination simulation are presented in Supplementary Table S5.

Due to the scope of the efficacy studies, it is not possible to estimate the rate of waning of vaccine-induced effects, $1/\rho$. Instead, for each main scenario, three options are explored with effects lasting one, three and ten years. The relative infectiousness of vaccinated infected animals and of vaccinated reinfected animals compared to unvaccinated naïve animals was parameterised using viral RNA shedding data[6], assuming that infectiousness is either proportional to viral RNA shedding or to the log of viral RNA shedding. Vaccination is implemented in an age dependent manner and occurs immediately at the point at which camels reach the age being targeted for vaccination. To evaluate the ideal age for vaccination under the model assumptions, the target age group was varied from one month old to four years old. The vaccine efficacy was not modelled to vary with age. In scenarios 1 and 2 vaccination was assumed to reduce the relative infectiousness of first-time infected animals ($r_v$) by the same amount as natural infection reduces viral shedding in reinfected individuals. The relative infectiousness of reinfected vaccinated individuals ($r_{inf\_v}$) was estimated as 0.15% when a linear relationship between infectiousness and viral RNA shedding was assumed, and 33% when infectiousness was assumed to be proportional to the log of viral RNA shedding[6]. See Table 4 for the complete set of parameters used to model the effect of vaccination. Vaccine impact was measured as difference in incidence following annual vaccination over a ten-year period and potential to disrupt patch or population-level transmission.

### Reporting summary

Further information on research design is available in the Nature Portfolio Reporting Summary linked to this article.

## Data availability

All data used in model fitting or parameterisation are available either within this published article or at https://doi.org/10.5281/zenodo.15864039[52].

## Code availability

All models presented in this manuscript and the code used to produce the analyses are available at: https://doi.org/10.5281/zenodo.15864039[52].

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

## Acknowledgements

The authors acknowledge helpful input from Maria Van Kerkhove in shaping the focus of this work. The authors acknowledge funding from the Wellcome Trust Studentship 203871/Z/16/Z (A.D.), the Medical Research Council (MRC) Centre for Global Infectious Disease Analysis (MR/X020258/1) funded by the UK MRC and carried out in the frame of the Global Health EDCTP3 Joint Undertaking supported by the EU (A.D., T.J., N.F.); the NIHR for support for the Health Research Protection Unit in Modelling and Health Economics, a partnership between the UK Health Security Agency (UKHSA), Imperial College London and London School of Hygiene & Tropical Medicine (grant code NIHR200908) (A.D., T.J., N.F.); a philanthropic donation from Community Jameel supporting the work of the Jameel Institute (A.D., T.J., N.F.). The funders had no role in the study design, data collection, data analysis, data interpretation, or writing of the report. For the purpose of open access, the author has applied a 'Creative Commons Attribution' (CC BY) licence to any Author Accepted Manuscript version arising from this submission.

## Author contributions

Conceptualisation: N.F., A.D. and T.J., formal analysis: A.D., methodology: N.F., A.D. and T.J., manuscript writing – original draft A.D., writing - review and editing N.F. and T.J.

## Competing interests

The authors declare no competing interests.
