## [Transparent Peer Review file · Nature Communications]

Modelling transmission of Middle East respiratory syndrome coronavirus in camel populations and the potential impact of animal vaccination

Corresponding Author: Dr Amy Dighe

Version 0:

Reviewer comments:

Reviewer #1

(Remarks to the Author)

The paper is clear and comprehensive. The topic and research is novel, and the authors are addressing issues and questions that are important knowledge gaps for MERS-CoV. Because MERS-CoV is a subclinical infection in camels, case data (case counts, incidence measure, etc...) are largely absent, leading to a huge challenge in understanding the epidemiology and transmission dynamics of the infection in the reservoir population. It means that researchers must rely on field serological data and controlled transmission studies, both of which have significant drawbacks. Because of this, there are uncertainties and unknowns in the data, leading the authors to make a number of assumptions and generalizations. These include:

- Immunity: Seropositivity is not necessarily an indication of protective immunity. It has been documented that camels may become infected multiple times, including in the presence of high levels of antibodies. Basing FOI and R0 on seroprevalence data introduces questions about the validity of the results.
- Relationship between viral shedding and infectiousness
- The effect of vaccination in field trials
- Periodicity or seasonality of infections
- Connectedness of different camel populations (camel mixing patterns)

While the authors did an excellent job of addressing the issues listed above (and others not listed), the data gaps unavoidably created uncertainty in the analyses, leading to wide intervals in the results and questions around the applicability in real-world scenarios. I agree with their conclusion, that their research provides a framework for simulating vaccination strategies based on data from contextual demographics and further empirical studies.

In addition to the above general comments, I have two questions to be addressed:

1/ In human population-based FOI estimates using age stratified seroprevalence data, the sampling data tend to have high resolution (many age categories with a width of a few months, and many data points for each age category). Since FOI was estimated on a per-study basis, with different sample sizes, age categories, etc... How does the lack of fine resolution of the camel demographic data affect the FOI models?

2/ Figure 5, table 4, line 417. Notation for FOI should be consistent. In Figure 5, λ_1 and $\phi \lambda_1$ are used, but ϕ is not defined in table 4. Line 417 refers to λ_2 but it is not defined.

(Remarks on code availability)

Reviewer #2

(Remarks to the Author)

The study presents an ambitious and methodologically sounding exploration of MERS-CoV transmission dynamics in camels and evaluates the potential impacts of vaccination as a mitigation strategy. While the research is methodologically robust and provides critical insights, it has several limitations that must be considered when interpreting its findings.

The use of age-stratified seroprevalence from published studies to model MERS-CoV transmission in camels is a significant advancement. The dynamic model allows for nuanced evaluations of vaccination scenarios, marking an important step forward in zoonotic disease modeling. However, the reliance on assumptions for parameters like seroreversion rates and the relationship between viral shedding and infectiousness introduces uncertainty. While sensitivity analyses are conducted, the lack of empirical validation limits the reliability of some findings, particularly regarding vaccine efficacy.

R0 estimates span a wide range (e.g., 3 to 34 in the Middle East), which may limit their practical utility. While variability reflects the influence of different assumptions, such broad ranges can make it challenging for policymakers to draw definitive conclusions. The study shows that R0 estimates are particularly sensitive to the assumed relative infectiousness of reinfected animals. This reliance on a parameter with considerable uncertainty undermines the confidence in R0 estimates for regions with high FOI.

The wide range of CCS values (1,000–70,000 camels) is informative but lacks a clear connection to real-world scenarios. For instance, how do typical camel herd sizes or movement patterns in affected regions compare to these thresholds? A stronger emphasis on practical implications would make the results more actionable.

The study employs robust transmission models to simulate the impact of vaccination under varying scenarios, including different levels of vaccine efficacy, target ages, and population sizes. The findings on the efficacy of vaccinating young calves and the role of high vaccine coverage in reducing transmission provide actionable insights for policymakers. However, the study underexplores other potential interventions, such as improved surveillance, biosecurity measures, or husbandry modifications. A more comprehensive analysis could provide a broader range of actionable recommendations. The study does not fully address logistical and operational barriers to achieving high vaccination coverage, particularly in remote or resource-limited settings. Reaching 80-90% coverage in large populations may be challenging, even if theoretically effective.

Although the study acknowledges the importance of human-camel interactions in zoonotic spillover, it does not deeply explore these dynamics. This limits its applicability to evaluating public health risks and designing integrated control strategies.

In summary, the study provides a solid foundation for understanding MERS-CoV transmission in camels and evaluating vaccination strategies. However, the reliance on incomplete data, assumptions in key parameters, and limited exploration of alternative control measures restrict the study's impact.

(Remarks on code availability)

Reviewer #3

(Remarks to the Author)

This manuscript provides an interesting modeling analysis of the potential impact of the use of a MERS vaccine in domestic camel populations. A particular strength of the study is the use of a number of age-specific seroprevalence datasets to estimate of the annual force of infection in different camel populations, which is then used to parameterize and calibrate the transmission model. The FOI and R0 estimates provide the most comprehensive assessment to-date of the geographic variation in transmission intensity with camel populations. The hypothetical vaccination modeling scenarios nicely incorporated existing knowledge about MERS infection dynamics in camels, while also exploring the sensitivity of vaccine impact to the multiple uncertainties regarding the duration of immunity, etc. I have several questions regarding the model structure:

Did you consider using density-dependent transmission rather than frequency-dependent transmission? Livestock diseases are often at least partially density-dependent.

In catalytic models 3 and 4 with waning of mAbs, the value of m_0 should be derivable from the FOI, rather than set as a fixed parameter estimated from the simpler model. Relatedly, I could not tell from the dynamic model description or Figure 5 how the fraction of births protected by mAbs was determined? Is this a fixed fraction or are births proportional to the current fraction of immune adults in the population $(I+R)/N$?

It wasn't clear how different overall population sizes were represented in the structured metapopulation. Were there a fixed

number of patches with patch sizes that varied, or were there varying numbers of patches each with a fixed patch size? It would be interesting to see how these different structures influenced the CCS.

A fixed daily mortality rate for adults is used that appears to represent fairly long-lived populations. In some places camels are raised for meat and the mean age of the camel population is presumably much younger; if the mortality rate was significantly higher in these populations how would this impact the CCS and vaccine recommendations?

In the R0 section of the methods the transmission intensity parameter is labeled 'b', should this be 'beta' as used in the later sections?

(Remarks on code availability)

The github repository includes a README file that provides instructions on recreating the analyses presented in the paper, as well as instructions on how to run the vaccine simulation model. I did not attempt to run the entire model process as the instructions say it would take about 1 week run on a laptop.

Version 1:

Reviewer comments:

Reviewer #2

(Remarks to the Author)

The authors have responded thoroughly to my comments, and the revised manuscript is clear and highly valuable. In particular, despite limitations arising from gaps in knowledge about certain aspects of transmission, this work represents a significant advancement in understanding the epidemiology and control of MERS-CoV. I recommend acceptance.

(Remarks on code availability)

Reviewer #3

(Remarks to the Author)

The authors have thoroughly addressed all of my comments and suggestions from the previous version of this manuscript. The revised version is well written and highlights the new findings while also adequately acknowledging the limitations of the study.

(Remarks on code availability)

made.

Dear reviewers,

Thank you very much for your valuable feedback please find our responses to each of your comments below.

EDITOR'S COMMENTS

For your information, the referees have the following expertise:

Referee 1: viral epidemiology, MERS-CoV, epidemiology of camel-borne viruses

Referee 2: infectious disease epidemiology, mathematical modelling

Referee 3: infectious disease epidemiology, mathematical modelling, viral infections in wildlife

As you will see from the reports, the referees find the work of interest; however, they have raised some important concerns that we must ask you to address before we can reach a final decision on publication. In particular, we ask that you address their comments regarding the limitations of this work due to uncertainty in the analyses; we also ask that you address Referee 3's comments relating to the metrics used for the analysis. Please ensure that all of the referees' comments are addressed in full.

REVIEWERS' COMMENTS

Reviewer #1 (Remarks to the Author):

The paper is clear and comprehensive. The topic and research is novel, and the authors are addressing issues and questions that are important knowledge gaps for MERS-CoV. Because MERS-CoV is a subclinical infection in camels, case data (case counts, incidence measure, etc...) are largely absent, leading to a huge challenge in understanding the epidemiology and transmission dynamics of the infection in the reservoir population. It means that researchers must rely on field serological data and controlled transmission studies, both of which have significant drawbacks. Because of this, there are uncertainties and unknowns in the data, leading the authors to make a number of assumptions and generalizations. These include:

- Immunity: Seropositivity is not necessarily an indication of protective immunity. It has been documented that camels may become infected multiple times, including in the presence of high levels of antibodies. Basing FOI and R0 on seroprevalence data introduces questions about the validity of the results.

Thank you for your feedback. We note that the reliability of estimates of the force of infection (FOI) derived from seroprevalence data does not depend on seropositivity being indicative of immunological protection – catalytic models effectively estimate the per-capita rate at which seronegative animals seroconvert. FOI estimates are sensitive to assumptions about maternal antibodies and seroreversion, which is why we examined these factors in the hierarchy of catalytic models examined. We have clarified this in the results in lines 68-71 of the updated manuscript with the following text:

“Catalytic models estimate the per-capita rate at which seronegative animals seroconvert. Whilst the reliability of estimates does not depend on seropositivity being immunologically protective, such models are sensitive to assumptions about maternally acquired antibodies (mAbs) and seroreversion and so we fitted four different models to examine these factors.”

However, while FOI estimates don't assume seropositivity is protective, translating FOI estimates into R0 estimates does require making additional assumptions about the susceptibility of seropositive animals to reinfection and the contribution of those animals to transmission. Although there is evidence that reinfection occurs and can happen rapidly in very high transmission intensity high stress environments such as densely populated market pens, other evidence would indicate that seropositivity is at least a correlate of protection: for example, in longitudinal calf-dam pair studies MERS-CoV infection has been observed to spread through calves shortly after seropositivity from maternally acquired antibodies have waned. In cross-sectional studies, evidence of active infection (e.g. as indicated by PCR positivity) is much more likely to be found in young animals than older (mostly seropositive) ones, again indicative of at least partially protective acquired immunity. Hence, we examined three scenarios for waning of complete immunity (none, after 90 days and after 30 days) and two scenarios of susceptibility to future infection following the waning of complete immunity (75% susceptibility and total susceptibility) into our transmission dynamic models, exploring two assumptions about the relative infectiousness of reinfections relative to primary infections. For each such immunity scenario, R0 values were calibrated to reproduce the range of FOI estimates obtained from the catalytic models used to fit seroprevalence data.

- Relationship between viral shedding and infectiousness
- The effect of vaccination in field trials
- Periodicity or seasonality of infections
- Connectedness of different camel populations (camel mixing patterns)

While the authors did an excellent job of addressing the issues listed above (and others not listed), the data gaps unavoidably created uncertainty in the analyses, leading to wide intervals in the results and questions around the applicability in real-world scenarios. I agree with their conclusion, that their research provides a framework for simulating vaccination strategies based on data from contextual demographics and further empirical studies.

Thank you, and we agree that there is unavoidable uncertainty stemming from data gaps that this work highlights, leading us to model several different possible vaccination scenarios. However, we feel that our work adds value given the prior absence of existing estimates of MERS-CoV transmissibility in camels. Despite the wide intervals around some of our estimates, there are important conclusions that can be drawn which could support real-world decision making and guide further research priorities, including: 1) MERS-CoV transmission intensity varies geographically, with higher FOI/R0 estimates in populations the Middle East compared to Africa and South Asia, 2) the critical community size for MERS-CoV in camels is larger than the size of most herds emphasising the importance of preventing herd-to-herd transmission and 3) that vaccinating calves against MERS-CoV could lead to large reductions in incidence in dromedaries, even if vaccines only reduce infectiousness rather than protecting individuals from infection.

In addition to the above general comments, I have two questions to be addressed:

1/ In human population-based FOI estimates using age stratified seroprevalence data, the sampling data tend to have high resolution (many age categories with a width of a few months, and many data points for each age category). Since FOI was estimated on a per-study basis, with different sample sizes, age

categories, etc... How does the lack of fine resolution of the camel demographic data affect the FOI models?

The reviewer is correct that if we had access to data for finer age strata, we might be able to reduce the uncertainty of some of our FOI estimates. Statistically, the resolution of age stratification is most important for ages below the mean age of first infection, which for MERS-CoV in camels is in the first few years of life. Limited additional power would be gained from finer stratification of age in adult animals (while many of the studies we used had wide strata or a single stratum for adult animals, seroprevalence approached 100% in these adults). Most strata for the youngest animals were 1-2 years wide providing sufficient resolution where age discrimination was most important, but we appreciate the point that there is variation between width of strata across studies. To further address the question of how the resolution of age strata could be affecting our results, we refitted our FOI (catalytic) models while aggregating the age strata in the two most finely stratified studies into 2 age categories and compared our estimates with the originals. We produced very similar FOI estimates using this aggregated data as we had with the finer resolution original data. For the study by Hemida et al in KSA we aggregated the four age strata (<1yr, 1-3yrs, 4-5yrs, >5yrs) into two (≤ 3 years and > 3 years) and estimated a FOI of 3.2/yr (95% CrI: 1.3, 8.3) which is similar to our original estimate of 3.0/yr (95% CrI: 1.7, 9.1). For the study by Saqib et al in Pakistan, we aggregated the four age strata (≤ 2 yrs, 2.1-5yrs, 5.1-10yrs, > 10 yrs) into two (≤ 5 years and > 5 years) and estimated an annual FOI of 0.1/yr (95% CrI: 0.1, 0.2), very similar to our original estimate. Given the young age at first infection for MERS-CoV in many camel populations we expect that more studies with strata < 1 year wide in young animals would likely improve our estimates but we would not expect the differences to change our conclusions about the controllability of MERS-CoV or the relative transmission intensity in the Middle East versus Africa and Asia.

2/ Figure 5, table 4, line 417. Notation for FOI should be consistent. In Figure 5, λ_1 and $\phi \lambda_1$ are used, but ϕ is not defined in table 4. Line 417 refers to λ_2 but it is not defined.

Thank you for catching this inconsistency. We have now updated the notation for the FOI in the table and the text to match the figure.

Reviewer #2 (Remarks to the Author):

The study presents an ambitious and methodologically sounding exploration of MERS-CoV transmission dynamics in camels and evaluates the potential impacts of vaccination as a mitigation strategy. While the research is methodologically robust and provides critical insights, it has several limitations that must be considered when interpreting its findings.

The use of age-stratified seroprevalence from published studies to model MERS-CoV transmission in camels is a significant advancement. The dynamic model allows for nuanced evaluations of vaccination scenarios, marking an important step forward in zoonotic disease modeling. However, the reliance on assumptions for parameters like seroreversion rates and the relationship between viral shedding and infectiousness introduces uncertainty. While sensitivity analyses are conducted, the lack of empirical validation limits the reliability of some findings, particularly regarding vaccine efficacy.

Thank you for your feedback. We appreciate that the data gaps for some quantities, such as the relationship between viral shedding and infectiousness, introduces uncertainty around our estimates and necessitates conducting extensive sensitivity analyses. However, given the lack of previous estimates of the transmissibility of MERS-CoV and the potential impact of vaccinating camels, we believe that our results and the framework we developed for simulating animal vaccination still add value and represent a step forward in our understanding of MERS-CoV in the animal reservoir.

R0 estimates span a wide range (e.g., 3 to 34 in the Middle East), which may limit their practical utility. While variability reflects the influence of different assumptions, such broad ranges can make it challenging for policymakers to draw definitive conclusions. The study shows that R0 estimates are particularly sensitive to the assumed relative infectiousness of reinfected animals. This reliance on a parameter with considerable uncertainty undermines the confidence in R0 estimates for regions with high FOI.

The R0 estimate of 34 is a major outlier for a single study in UAE in which the majority of very young calves were seropositive suggesting there had recently been a large outbreak in the calf population. Ignoring this one study the range in the Middle east is 3-17. The overall interquartile range of estimates in the Middle east is 7-14 so perhaps this is a more meaningful measure, that better represents the overall distribution, to present and to compare across regions that might have more practical utility. We have now updated the text both in the abstract and the main text in the Results section line 109 to present the interquartile range as a better representation of the general distribution of R0 estimates in each region.

With regard to the sensitivity of R0 estimates to the occurrence of and relative infectiousness of reinfections, infectiousness is often assumed to be linearly related to viral load, as we assume in our central analysis. This leads to an estimate of reinfections being 1% as infectious as primary infections. However, to illustrate the extent to which uncertainty around reinfection affects R0 estimates (and thus the predicted impact of vaccination on transmission) we also examined an extreme alternative where infectiousness is proportional to the logarithm of viral load, leading to reinfections being 50% as infectious as primary infections. We note that R0 estimates are highest (and thus vaccination has the least impact on transmission) when infectiousness of reinfected animals is assumed to be low compared with primary infections. We have added this text to the discussion of limitations (lines 290-293):

“Last, we note that R_0 estimates are sensitive to uncertainty about the relative infectiousness of reinfections. Estimates are highest (and thus vaccination has the least impact on transmission) when infectiousness is assumed to be linearly related to viral load, leading to low infectiousness of reinfected animals compared to primary infections.

We have also described this sensitivity more clearly in the results on lines 110-115:

“These estimates were based on the common assumption that infectiousness is linearly related to viral load, which leads to an estimate of reinfections being 1% as infectious as primary infections based on shedding data from the control arm of the ChAdOx vaccine field study⁶. However, to illustrate the extent to which uncertainty around reinfection affects R_0 estimates, we also examined an extreme alternative where infectiousness is proportional to the logarithm of viral load leading to reinfections being 50% as infectious as primary infections (Table 1).”

The wide range of CCS values (1,000–70,000 camels) is informative but lacks a clear connection to real-world scenarios. For instance, how do typical camel herd sizes or movement patterns in affected regions compare to these thresholds? A stronger emphasis on practical implications would make the results more actionable.

Under our central assumptions the CCS was estimated to be between 11,600-67,200 depending on the transmission setting, degree of seasonality of calving and underlying herd structure – these factors are all likely to vary across settings and so the true CCS is also likely to vary across settings. Unlike in the UK (post BSE/FMD), there are limited data available on herd sizes and movement patterns in affected regions, which hinders our ability to improve model realism. Data from Qassim in KSA suggest that herd sizes are mostly well under 1000 with exceptions for intensive dairy farms. We have this text in the discussion “Our estimates of the CCS were larger than most reported herd sizes, which tend to be well under 1,000 animals^{33,34,36}, emphasising the importance of focusing interventions on reducing inter-herd infections for interrupting transmission.” And have now added an emphasis on the practical implications in the Results section on lines 127-130 to include “The CCS was estimated to be well above reported herd sizes³¹⁻³³, and varied between approximately 10,000 – 70,000 camels depending on the transmissibility, seasonality of calving and underlying herd structure assumed in the transmission model (Table 2, Figure 2)”.

The study employs robust transmission models to simulate the impact of vaccination under varying scenarios, including different levels of vaccine efficacy, target ages, and population sizes. The findings on the efficacy of vaccinating young calves and the role of high vaccine coverage in reducing transmission provide actionable insights for policymakers. However, the study underexplores other potential interventions, such as improved surveillance, biosecurity measures, or husbandry modifications. A more comprehensive analysis could provide a broader range of actionable recommendations. The study does not fully address logistical and operational barriers to achieving high vaccination coverage, particularly in remote or resource-limited settings. Reaching 80-90% coverage in large populations may be challenging, even if theoretically effective.

We did not include these wider intervention measures in this modelling study, as data do not exist to allow us to parameterise the impacts of enhanced biosecurity or husbandry modifications within our relatively simple transmission model structure. Enhanced surveillance is relevant for reactive control measures (e.g. depopulation of or trading restrictions placed on infected herds), but modelling such reactive control policies would have considerably extended the scope of what is already a long paper. However, we completely agree that non-vaccine based interventions will remain critical to control. We have added text to the discussion lines 325-327 “. It should be noted that vaccination of dromedaries is not likely to be used in isolation but rather considered as part of a suite of complimentary control measures in the animal population. However, to meaningfully consider interventions such as biosecurity measures or potential changes to husbandry practices in this modelling analysis would require much more detailed data on current husbandry practices and population structures.”

Although the study acknowledges the importance of human-camel interactions in zoonotic spillover, it does not deeply explore these dynamics. This limits its applicability to evaluating public health risks and designing integrated control strategies.

We agree that this is an important next step for facilitating design and cost-effectiveness analyses of integrated control strategies. Characterising the interface between camels and people and extending this framework to include spillover dynamics is a research priority that we are particularly interested in - but one we feel is beyond the scope of the current paper.

In summary, the study provides a solid foundation for understanding MERS-CoV transmission in camels and evaluating vaccination strategies. However, the reliance on incomplete data, assumptions in key parameters, and limited exploration of alternative control measures restrict the study's impact.

Thank you for helping us frame our results in clearer and more impactful way. Whilst we appreciate that data gaps meant we needed to consider multiple potential parameter values in some cases, we have still been able to reach some impactful conclusions which could support decision making and guide further research priorities, including: 1) the geographical variation in MERS-CoV transmission intensity, with higher FOI/R0 estimates in populations in the Middle East compared to Africa and South Asia, 2) that the critical community size for MERS-CoV in camels is larger than the size of most herds, emphasising the importance of preventing herd-to-herd transmission and 3) that vaccinating calves against MERS-CoV could lead to large reductions in incidence in dromedaries, even if vaccines only reduce infectiousness rather than protecting individuals from infection. We agree that it will be important for future work to extend our current analysis to examine the potential impact of additional control measures for MERS-CoV control.

Reviewer #3 (Remarks to the Author)

This manuscript provides an interesting modeling analysis of the potential impact of the use of a MERS vaccine in domestic camel populations. A particular strength of the study is the use of a number of age-specific seroprevalence datasets to estimate of the annual force of infection in different camel populations, which is then used to parameterize and calibrate the transmission model. The FOI and R0 estimates provide the most comprehensive assessment to-date of the geographic variation in transmission intensity with camel populations. The hypothetical vaccination modeling scenarios nicely incorporated existing knowledge about MERS infection dynamics in camels, while also exploring the sensitivity of vaccine impact to the multiple uncertainties regarding the duration of immunity, etc. I have several questions regarding the model structure:

Did you consider using density-dependent transmission rather than frequency-dependent transmission? Livestock diseases are often at least partially density-dependent.

Thank you for this important question. We agree that it is plausible that there is some density-dependence, but we chose to model frequency-dependent transmission for a couple of key reasons. Firstly, we wanted to consider transmission over a wide range of scales/population sizes without assuming implicitly that these populations would all occupy the same sized area (for example when estimating the CCS). To meaningfully incorporate density-dependence would have required detailed data on the relative geographic sizes of camel farms comparable to that available for cattle in the UK (post BSE/FMD), but such data are just not available for camel populations currently. Secondly, in many countries camel farming is still overwhelmingly extensive and the areas that herds live/move within are large and change over time. In these extensive systems, transmission is likely to depend more on the proportion of individuals infected rather than the absolute number. However, to your point, as camel

farming becomes increasingly more intensive in parts of the Arabian Peninsula, it may well be that in some husbandry systems density dependent transmission would be a more appropriate model. This potential density-dependence could contribute to the variation in our transmissibility estimates across settings. We have added the following text to the methods to address this lines 414-420

“We chose to model transmission as frequency-dependent rather than density-dependent. In many countries camel farming is still overwhelmingly extensive³⁶, with herds living and moving within large areas. In such settings, transmission is likely to depend more on the proportion of individuals infected rather than the absolute number. Modelling frequency dependence also enabled us to consider transmission at a wide range of scales and population sizes without requiring detailed data on animal stocking densities. To do this in a meaningful way whilst incorporating density-dependence would have required additional data on camel densities of camel farms, grazing areas and markets and movement between these settings that is not currently available”.

And the following to the limitations of our transmissibility estimates in the discussion lines 281-283:

“Third, without high resolution data on camel population density and movements, we were not able to meaningfully explore potential density-dependence of transmission which could contribute to some of the variation in R0. Detailed data describing local camel populations could be used to tailor our models to specific settings and account for such dynamics.”.

In catalytic models 3 and 4 with waning of mAbs, the value of m_0 should be derivable from the FOI, rather than set as a fixed parameter estimated from the simpler model. Relatedly, I could not tell from the dynamic model description or Figure 5 how the fraction of births protected by mAbs was determined? Is this a fixed fraction or are births proportional to the current fraction of immune adults in the population $(I+R)/N$?

Thank you for highlighting this oversight. As you suggested, we have now derived the m_0 as laid out below for model 4 (also added to the Supplementary Materials, Equation S5).

For model 4, which includes maternally acquired antibodies that wane at rate ω and seroreversion at rate σ , the proportion of animals seropositive at age a is

$$p(a) = \frac{\lambda}{\lambda + \sigma} (1 - e^{-(\lambda+\sigma)a}) - m_0 \frac{\lambda}{\lambda + \sigma - \omega} (e^{-\omega a} - e^{-(\lambda+\sigma)a}) \quad S4$$

Noting that $\mu e^{-\mu(a-4)}$ is the age distribution of animals over 4 years of age, m_0 is given by

$$\begin{aligned} m_0 &= \int_4^{\infty} \mu e^{-\mu(a-4)} p(a) da \\ &= \int_4^{\infty} \frac{\lambda \mu e^{4\mu}}{\lambda + \sigma} (e^{-\mu a} - e^{-(\lambda+\sigma+\mu)a}) - m_0 \frac{\lambda \mu e^{4\mu}}{\lambda + \sigma - \omega} (e^{-(\omega+\mu)a} - e^{-(\lambda+\sigma+\mu)a}) da \\ &= \frac{\lambda}{\lambda + \sigma} \left[1 - \frac{\mu}{\lambda + \sigma + \mu} e^{-4(\lambda+\sigma)} \right] - m_0 \frac{\lambda \mu}{\lambda + \sigma - \omega} \left[\frac{e^{-4\omega}}{\omega + \mu} - \frac{e^{-4(\lambda+\sigma)}}{\lambda + \sigma + \mu} \right] \end{aligned}$$

So

$$m_0 = \frac{\frac{\lambda}{\lambda + \sigma} \left[1 - \frac{\mu}{\lambda + \sigma + \mu} e^{-4(\lambda + \sigma)} \right]}{1 + \frac{\lambda \mu}{\lambda + \sigma - \omega} \left[\frac{e^{-4\omega}}{\omega + \mu} - \frac{e^{-4(\lambda + \sigma)}}{\lambda + \sigma + \mu} \right]}$$

We have updated our code in the git hub repository, and our estimates in Table 1 to reflect this change in the model. The new estimates are very similar to the originals (the great majority of central estimates for the FOI are the same as the originals to 1 decimal place with only 4 changing by +/- 0.1 and so the downstream modelling work was not affected. Thank you again for catching this and allowing us to improve the model.

With regard to the second part of your question about how the fraction of births protected by mAbs was determined in the dynamic model, it is indeed based off of the proportion of adults >4 years of age that are immune. We have now added this to the Figure 5 to make it clearer (copy pasted below).

It wasn't clear how different overall population sizes were represented in the structured metapopulation. Were there a fixed number of patches with patch sizes that varied, or were there varying numbers of patches each with a fixed patch size? It would be interesting to see how these different structures influenced the CCS.

In our central results there were a fixed number of patches of varying sizes (25 in a 5x5 grid) rather than a fixed patch size with varying number of patches. We made this decision based on the practicalities of coding the model and the run time. However, we appreciate the importance of the point you raise and to address the influence of different patch structures on the CCS we reran our analysis at 2 additional alternative patch structures – 9 patches in a 3x3 grid and 49 patches in a 7x7 grid. In the figure below, we see that the total population size is more important than the number of patches for the percentage of runs in which transmission persists. Comparing, e.g., a total population size of 75,000 split into 49 patches (each of ~1530 camels) to a total population size of 40,000 split into 25 patches (each of 1600 camels), there is still greater persistence in the larger population than the smaller population made up of the same size patches. But the number and the size of the patches does influence the persistence, with more smaller patches meaning lower persistence and therefore a larger CCS (except for high R_0 scenarios), with a larger total population size needed to sustain transmission. We have now better clarified how the different population sizes were represented in the methods and results and refer to the figure below which we have added to the supplementary material as Figure S6.

Figure S6. The effect of the degree of population structure on the percentage of stochastic model runs in which transmission persists for at least 25 years. Persistence is shown for a given total population size (x axis) broken down into increasing number of smaller sub-populations (y axis), for three different transmission intensities. At low ($R_0 = 3.5$) and moderate ($R_0 = 7$) transmission intensities the increasing structure in the population reduces persistence, whereas at a high transmission intensity ($R_0 = 14$) the

increasing structure increases persistence. This analysis was run assuming reinfections are 1% as infections as reinfections and births reflect the full degree of seasonality representative of calving in Kingdom of Saudi Arabia.

Added to methods lines 499-502:

“For our central results we fixed the number of patches at 25, varying the patch size to represent different total population sizes. We conducted a sensitivity analysis to understand the effect patch size had on transmission persistence varying the number of patches from 9 (a 3x3 grid) to 49 (a 7x7 grid).”

Added to results lines 136-140:

“Although sustained transmission was most dependent on total population size, the patch size did influence persistence. For low and moderate R_0 values, the greater the number of patches a given total population was divided into, the less likely transmission was to be sustained. However, for high R_0 values, increasing the number of patches enhanced persistence (Figure S6) – likely due to the so-called “rescue effect”, whereby, when transmission dies out in a local patch, infection may be reintroduced from neighbouring patches with asynchronous epidemics³⁴”

A fixed daily mortality rate for adults is used that appears to represent fairly long-lived populations. In some places camels are raised for meat and the mean age of the camel population is presumably much younger; if the mortality rate was significantly higher in these populations how would this impact the CCS and vaccine recommendations?

The fixed mortality rates used equate to ~40% mortality per year in the first 2 years of life and ~15% after that. This equates to an average life expectancy of 4.9 years. At zoographic equilibrium we end up with an average of:

- *16% of animals between 0-1 year old*
- *12% 1-2 years*
- *9% 2-3 years*
- *7% 3-4 years*
- *56% >4 years. (we do not breakdown adult camels >4 years into smaller age categories since infection dynamics and vaccine targeting would not likely vary by age among adults).*

This age structure is in line with published data on camel zoography in the major camel farming region of Saudi Arabia. Looking at the age pyramid published by Abbas et al, and adding across males and females the total percentages in these age classes are:

- *14% are 0-1 year*
- *12% are 1-2 years*
- *10% 2-3 years*
- *9% in 3-4 years*
- *55% >4 years*

[REDACTED]

Source: Abbas, B., Al-Qarawi, A.A. and Al-Hawas, A. (2000) Survey on camel husbandry in Qassim region, Saudi Arabia: Herding strategies, productivity and mortality. Revue d'Élevage et de Médecine Vétérinaire des Pays Tropicaux, 53, 293-298.

Camels reach reproductive maturity at around 4 years of age and live about 20-25 years if allowed but, as you mentioned, many farmed camels die younger either due to being killed for meat (largely males 2-3 years old – meat from adult animals is very tough and not as flavourful), or due to disease/accidents. Based on our mortality rates, <2 % of all camels in the model live longer than 21 years which is also fairly similar to the findings of Abbas et al.

There is a market for camel meat, particularly in the Middle East where it is mostly imported from Africa but also increasingly in parts of Africa itself. However, in both Africa and the Middle East camels are overwhelmingly not raised specifically for meat. Rather, camel owners/herders tend to retain females in their herds for milk and send most young males to slaughter (some for meat, but often they are just culled) or sell them to be used as pack animals. This is certainly what we have gathered in discussions with those involved in the industry and from available descriptions in the literature. For example Mirkena et al say for Ethiopia:

“ The majority of camels slaughtered are culls, and only a limited number of castrated males are especially raised for slaughter. Camel meat markets and camel meat consumption are, with the exception of Sudan, not very well developed in Eastern Africa, but lucrative export opportunities to Egypt, Libya, Saudi Arabia and the Gulf States do exist. Due to the intrinsically low reproductive rate, camels are not efficient meat producers. Offtake rates of 3 to 5% might already constitute a stress on the population (Schwartz and Walsh 1992). Although many pastoralists consume camel meat when available, camels are never slaughtered for home consumption of meat except occasionally during festive times, to fulfil cultural obligations such as funerals, wedding ceremonies and religious festivals or when camels are accidentally injured (Wolde 1991; Eyasu 2009).”

Source: Mirkena, T., Walelign, E., Tewolde, N. et al. Camel production systems in Ethiopia: a review of literature with notes on MERS-CoV risk factors. *Pastoralism* 8, 30 (2018).
<https://doi.org/10.1186/s13570-018-0135-3>

Similar descriptions exist for camel husbandry in Kenya and Sudan. More intensive farming is beginning to exist in parts of Africa and is established in the Middle East in the form of dairy farms largely made up of adult (>4 years) females, rather than meat farms which would have a high turnover and low average age.

In summary, our high calf mortality and lower mortality rates in adulthood reflect the killing of male calves as part of mixed herds as described in the literature characterising dominant husbandry practices in both the Middle East and Africa. We do not think that a higher mortality rate in adults would be representative of existing camel husbandry systems. However, the industry is changing and intensifying rapidly – if feedlots of young male camels exclusively raised for meat became a major part of the husbandry systems we would expect that to have implications for vaccination impact on such farms. To fully address the point you raise we modelled an extreme scenario with annual adult mortality of 40%, giving a mean life expectancy of ~2.5 years. We observed the CCS to decrease in most scenarios, but to increase when the population is assumed to be homogenous and the birth seasonality reflects that described in KSA. This is likely to be due to complex dynamics around the availability of susceptible camels, with the epidemic being able to spread rapidly through the young population, driving itself extinct. With a high mortality of 40% in adults, a higher vaccine coverage was required to achieve the same reductions in incidence or to interrupt transmission than when mortality was representative of what we see in the literature. The coverage needed to interrupt transmission in low transmission intensity settings varied from 50-80% dependent on the length vaccine effects lasted, whereas only 40% coverage was required in our original analysis. In high transmission intensity settings, there was no little difference in the coverage required to interrupt transmission when we increased adult mortality, likely because the infections are mostly happening in the very young animals in these settings.

We have added a point about dependence on mortality rates to the results lines 240-245:

“These estimates are based on mortality rates that reflect the current dominant camel husbandry systems described in Eastern Africa and Arabian Peninsula in which calves experience a high mortality rate largely driven by the slaughter of young males and then surviving adults (mostly females) experience a lower mortality rate. If meat production becomes mostly intensive in the future, characterised by large dense farms with rapid turnover of animals, we would expect a higher vaccine coverage to be necessary to interrupt transmission in these settings. For example, if adult mortality were to rise to be comparable to calf mortality at approximately 40% annually, the vaccine coverage needed to interrupt transmission would increase from <50% coverage to between 50-80% coverage in a low transmission setting, depending on the duration of vaccine impact.

In the R0 section of the methods the transmission intensity parameter is labelled 'b', should this be 'beta' as used in the later sections?

In the manuscript as we see it, the parameter is labelled 'β' throughout. However, I wonder if in the version shared with reviewers it appears differently since the symbol font was used. We have gone through and inserted beta as a symbol instead of using symbol font – hopefully this has fixed the issue. Thanks for bringing this to our attention!

Reviewer #3 (Remarks on code availability):

The github repository includes a README file that provides instructions on recreating the analyses presented in the paper, as well as instructions on how to run the vaccine simulation model. I did not attempt to run the entire model process as the instructions say it would take about 1 week run on a laptop.

Thank you very much for your thorough comments and questions. We really appreciate the time you have taken to think about how we could improve this manuscript and feel that we have ended up writing a better and more robust paper by acting on your feedback.